# FOURIERFLOW: FREQUENCY-AWARE FLOW MATCHING FOR GENERATIVE TURBULENCE MODELING

## ABSTRACT

Modeling complex fluid systems, especially turbulence governed by partial differential equations (PDEs), remains a fundamental challenge in science and engineering. Recently, diffusion-based generative models have gained attention as a powerful approach for these tasks, owing to their capacity to capture long-range dependencies and recover hierarchical structures. However, we present both empirical and theoretical evidence showing that generative models struggle with significant spectral bias and common-mode noise when generating high-fidelity turbulent flows. Here we propose FourierFlow, a novel generative turbulence modeling framework that enhances the frequency-aware learning by both implicitly and explicitly mitigating spectral bias and common-mode noise. FourierFlow comprises three key innovations. Firstly, we adopt a dual-branch backbone architecture, consisting of a salient flow attention branch with local-global awareness to focus on sensitive turbulence areas. Secondly, we introduce a frequency-guided Fourier mixing branch, which is integrated via an adaptive fusion strategy to explicitly mitigate spectral bias in the generative model. Thirdly, we leverage the high-frequency modeling capabilities of the masked auto-encoder pre-training and implicitly align the features of the generative model toward high-frequency components. We validate the effectiveness of FourierFlow on three canonical turbulent flow scenarios, demonstrating superior performance compared to state-of-the-art methods. Furthermore, we show that our model exhibits strong generalization capabilities in challenging settings such as out-of-distribution domains, long-term temporal extrapolation, and robustness to noisy inputs. The code can be found at https://anonymous.4open.science/r/FourierFlow-847D.

## 1 INTRODUCTION

Diffusion and flow models (Song & Ermon, 2019; Ho et al., 2020; Lipman et al., 2023) have emerged as a powerful framework for generating high-fidelity data. By iteratively denoising noisy inputs, they can effectively capture both the global structures and fine-grained details of complex data, demonstrating clear advantages not only in modeling visual signals (Rombach et al., 2022; Ho et al., 2022) but also in fluid dynamics (Huang et al., 2024; Oommen et al., 2024; Wang et al., 2024). A recent study (Khodakarami et al., 2025) further highlights that diffusion model outperforms the standard Neural Operator (NO) by inherently learning to perturb and reconstruct signals across multiple scales, rather than operating on a fixed scale throughout training, as NO does. It enables a generative model to more faithfully recover the hierarchical structures of the underlying fluid systems. Given the intrinsically multi-scale and high-dimensional nature, generative model emerges as a promising paradigm for simulating fluid flow.

However, despite recent advances, generative models have not yet been widely adopted for turbulence modeling tasks. Rapid variations in local regions of turbulence typically correspond to significant fluctuations in the high-frequency components of the signal spectrum. This naturally raises an important question: ***Can standard generative model achieve high-fidelity turbulence modeling, providing scalability and generalization capabilities across diverse flow regimes?*** We believe the answer is negative due to fundamental limitations in two respects.

**(1) Spectral bias.** We provide two strong pieces of evidence. *From an empirical perspective*, we observe through extensive spectral analysis that these models tend to underrepresent high-frequency and high-energy components in Figure 1, which are critical for accurately reconstructing fine-scale

turbulent structures such as vortices, shocks, and shear layers. *From a theoretical standpoint*, this phenomenon can be attributed to the intrinsic dynamics of the generation process. Specifically, during the forward process, high-frequency components in the input data are corrupted more severely and earlier than low-frequency components due to their smaller signal-to-noise ratios. As a result, the reverse denoising process, which typically proceeds from low to high frequencies, inherently favors reconstruction of coarse-scale structures first.

(2) **Common-mode noise in attention.** Differential attention (Ye et al., 2025) has shown that attention mechanisms can be affected by noise, often attending to irrelevant contextual background. More critically, highly nonlinear and high-frequency variations in localized turbulence regions. These small-scale yet dynamically critical structures are frequently averaged out or diluted at the global level, becoming what we refer to as common-mode components (Laplante et al., 2018). Therefore, existing generative models suffer from the slow evolution of the global flow field, which can obscure strongly dynamic local structures such as vortices and shear layers.

Moreover, turbulence modeling differs substantially from natural image generation in terms of structural requirements. While images may tolerate perceptual smoothing, fluid dynamics demands strict preservation of energy across scales to maintain physical consistency. In this context, the inability of diffusion models to faithfully reconstruct high-frequency modes not only reduces visual fidelity but also undermines physical plausibility and generalization.

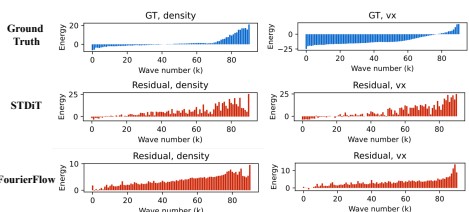

Figure 1: Demonstration of *spectral bias*. Residual is the gap between the prediction and the ground truth. Energy denotes the logarithm of the spectral energy, while the wavenumber indicates the frequency scale in the spectral domain. STDiT (Peng et al., 2025) fits the low-frequency components well but exhibits larger errors in the high-frequency range, resulting in residual spectra that are heavily concentrated in the high-wavenumber region. In contrast, our FourierFlow produces a more balanced residual spectrum by reducing spectral bias.

To overcome the limitations, we propose **Fourier-Flow**, a novel frequency-aware generative framework for turbulence modeling. FourierFlow incorporates both architectural and optimization innovations to explicitly and implicitly address spectral bias and common-mode noise. Specifically, **(1)** we design a novel Salient Flow Attention (SFA) mechanism to capture local-global spatial dependencies while suppressing irrelevant background signals. **(2)** We design a Frequency-guided Fourier Mixing (FFM) branch, forming a dual-branch backbone architecture to enhance the model's capacity for high-frequency components. These branches are fused adaptively to facilitate dynamic frequency signal integration. **(3)** On the optimization side, we leverage a pre-trained Masked Autoencoder (MAE) surrogate model to guide the generative model's alignment. Specifically, we match the intermediate representations of the generative model with those of a pre-trained MAE that is more sensitive to high-frequency features, thereby encouraging the generator to recover finer-scale structures.

Our empirical evaluation spans three canonical turbulence scenarios across both compressible and incompressible N-S flows. FourierFlow consistently outperforms state-of-the-art baselines in terms of accuracy, physical consistency, and frequency reconstruction. Notably, we demonstrate the model's strong generalization capabilities under challenging settings such as long-horizon temporal extrapolation, out-of-distribution flow regimes, and noisy observations.

## 2 PRELIMINARY

### 2.1 PROBLEM FORMULATION

Turbulent flow is always modeled by a governing PDE equation. We consider a PDE on $[0, T] \times \mathcal{D} \subset \mathbb{R} \times \mathbb{R}^d$ with the following form:

$$\frac{\partial u}{\partial t} = \mathcal{F}\left(u, \frac{\partial u}{\partial x}, \frac{\partial^2 u}{\partial x^2}, \dots\right) + f(\tau, x), \quad (\tau, x) \in [0, T] \times \mathcal{D}, \tag{1}$$

$$u(0, x) = u_0(x), \quad x \in D, \qquad B[u](\tau, x) = 0, \quad (\tau, x) \in [0, T] \times \partial\mathcal{D},$$

where $u : [0, T] \times \mathcal{D} \to \mathbb{R}^n$ is the solution, with the initial condition $u_0(x)$ at time $t = 0$ and boundary condition $B[u](\tau, x) = 0$ on the boundary. $\mathcal{F}$ is a function and $f(\tau, x)$ is the force term.

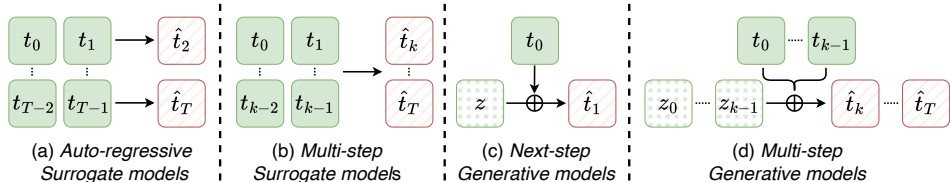

Figure 2: Comparison of existing architectures for turbulence modeling.

Let $u(\tau, \cdot) \in \mathbb{R}^{d \times n}$ denote the state at time step $t$. We are implementing multi-step generative modeling shown in Figure 2. Formally, an initial noisy trajectory of length $k$ is denoted as:

$$\mathcal{U}_0^k = \{u_0, u_1, \ldots, u_{k-1}\}, \quad u_\tau \sim p(u_\tau \mid u_{<\tau}, \boldsymbol{\epsilon}_\tau),$$

where $\boldsymbol{\epsilon}_\tau \sim \mathcal{N}(0, \sigma_\tau^2 I)$ represents stochastic perturbations capturing turbulence-induced uncertainty. To simulate the future evolution of the turbulent system, we aim to generate a predictive trajectory $\hat{\mathcal{U}}_k^{2k} = \{\hat{u}_k, \hat{u}_{k+1}, \ldots, \hat{u}_{2k-1}\}$ directly from Gaussian noise: $\hat{\mathcal{U}}_k^{2k} = \mathcal{G}_\theta(\boldsymbol{\epsilon}_{k:2k-1}, \mathcal{U}_0^k)$, where $\boldsymbol{\epsilon}_{k:2k-1} = \{\boldsymbol{\epsilon}_k, \ldots, \boldsymbol{\epsilon}_{2k-1}\}$ with $\boldsymbol{\epsilon}_\tau \sim \mathcal{N}(0, \sigma_\tau^2 I)$, and $\mathcal{G}_\theta$ is a learnable generative model parameterized by $\theta$, conditioned on the observed past $\mathcal{U}_0^k$. $k$ is set to 4 in our paper.

## 2.2 COMMON-MODE NOISE

*Common-mode noise* is the component of $n \in \mathbb{R}^C$ shared across channels, lying in $\mathcal{S}_{\text{cm}} = \text{span}\{\mathbf{1}_C\}$ with projector $P_{\text{cm}} = \frac{1}{C}\mathbf{1}_C\mathbf{1}_C^\top$ and complement $P_{\text{df}} = I_C - P_{\text{cm}}$. Thus $n = n_{\text{cm}} + n_{\text{df}}$ with $n_{\text{cm}} = P_{\text{cm}}n = \alpha\mathbf{1}_C$ and $\alpha = \frac{1}{C}\mathbf{1}_C^\top n$, so that the variance splits orthogonally: $\mathbb{E}\|n\|_2^2 = \mathbb{E}\|n_{\text{cm}}\|_2^2 + \mathbb{E}\|n_{\text{df}}\|_2^2$. For a prediction residual $e = \hat{u} - u$, we estimate its common-mode part as $\hat{e}_{\text{cm}} = P_{\text{cm}}e$ and penalize it via $\mathcal{L}_{\text{cm}} = \lambda_{\text{cm}}\int\|\hat{e}_{\text{cm}}\|_2^2$. A frequency-selective variant $\mathcal{L}_{\text{cm}}^{\text{freq}} = \mu_{\text{cm}}\sum_{k \in K_{\text{low}}}\|P_{\text{cm}}\widehat{E}(k, \cdot)\|_2^2$ targets spatially coherent, low-frequency drifts.

In attention mechanisms, adding $n_{\text{cm}}$ to all tokens shifts $Q, K$ by rank-1 terms, yielding an additive bias $QK^\top \mapsto QK^\top + \beta\,\mathbf{1}\mathbf{1}^\top$ that flattens the softmax distribution and suppresses token discrimination. Regularizing $\hat{e}_{\text{cm}}$ thus improves *contrastive sharpness* in attention maps, mitigating the tendency of attention weights to collapse toward uniform distributions under shared noise.

## 2.3 FLOW-BASED GENERATION VIA CONDITIONAL FLOW MATCHING

Flow matching (Lipman et al., 2023) formulates generative modeling as learning a continuous-time transport map between a simple base distribution $p_0$ (e.g., Gaussian noise) and the target data distribution $p_{\text{data}}$. Specifically, it parameterizes a time-dependent velocity field $\mathbf{v}_\theta(\mathbf{x}, t)$ and solves the deterministic ordinary differential equation

$$\frac{d\mathbf{x}(t)}{dt} = \mathbf{v}_\theta(\mathbf{x}(t), t), \quad \mathbf{x}(0) \sim p_0, \quad \mathbf{x}(1) \sim p_{\text{data}}, \tag{2}$$

so that the trajectory $\mathbf{x}(t)$ transports samples from $p_0$ to $p_{\text{data}}$. Training proceeds by matching $\mathbf{v}_\theta$ to a reference velocity $\mathbf{v}^*(\mathbf{x}, t)$ computed from an interpolation between base and data samples:

$$\mathcal{L}_{\text{flow}} = \mathbb{E}_{t \sim \mathcal{U}(0,1), \mathbf{x}(t)}\left[\|\mathbf{v}_\theta(\mathbf{x}(t), t) - \mathbf{v}^*(\mathbf{x}(t), t)\|^2\right], \tag{3}$$

where $\mathbf{x}(t) = (1 - t)\mathbf{x}_0 + t\mathbf{x}_1$, $\mathbf{x}_0 \sim p_0$, and $\mathbf{x}_1 \sim p_{\text{data}}$. Compared to discrete-step diffusion models, flow matching provides a *deterministic, non-iterative* sample generation process, offering faster inference and continuous-time control. These properties make it particularly appealing for physics-informed generative modeling, where fine-grained trajectory alignment and computational efficiency are critical.

# 3 METHOD

## 3.1 MOTIVATION

Our analysis reveals that even the most state-of-the-art generative models, whether in video generation or fluid simulation, suffer from spectral bias and common-mode noise. To address the challenges, we draw inspiration from differential attention (Ye et al., 2025) and propose a novel local-global

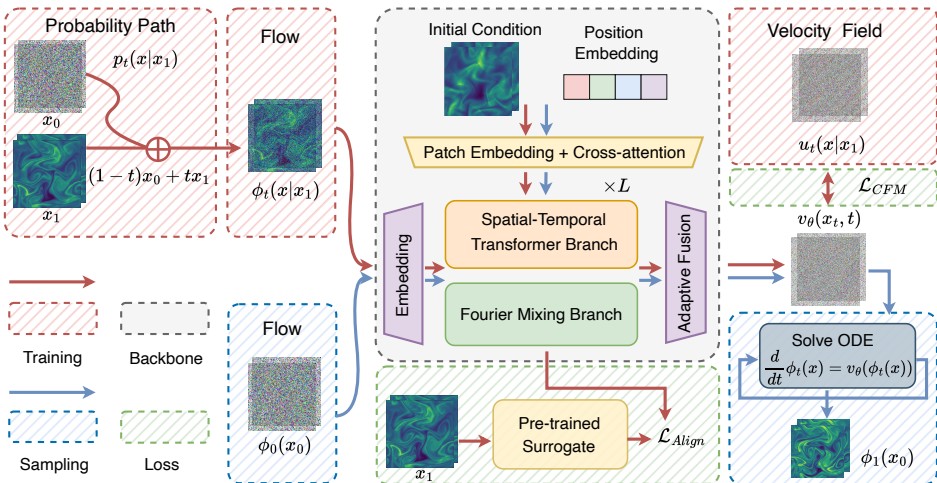

Figure 3: Overview of our proposed FourierFlow framework. We mark the training and sampling process in red and blue backgrounds, respectively.

differential attention mechanism (*i.e.,* SFA), which explicitly suppresses common-mode signals by emphasizing relative variations across spatial regions.

To mitigate spectral bias, we adopt both explicit and implicit strategies. The explicit approach is realized through the Fourier Mixing (FM) branch, which amplifies high-frequency feature extraction and adaptively fuses it with complementary representations. The implicit strategy leverages a pretrained surrogate model to guide the generative model via feature alignment. Specifically, we align intermediate representations with those from a frequency-sensitive pretrained Masked Autoencoder (MAE), encouraging the generator to preserve fine-scale structures.

## 3.2 ARCHITECTURE BACKBONE

The main dual-branch flow matching backbone is illustrated in Figure 3.

**Salient Flow Attention (SFA) Branch.** Differential attention (Ye et al., 2025) is motivated to amplify attention to the relevant context while canceling noise. Its core principle lies in the use of noise-canceling headphones and differential amplifiers to reduce common-mode noise (Laplante et al., 2018). Actually, the common-mode noise also exists in turbulence modeling in that highly nonlinear and high-frequency variations often lie in the relative variations between neighboring regions. For example, vorticity, a key descriptor of rotational flow, is defined as the spatial differential of the velocity field. Besides, velocity or pressure magnitudes can vary significantly across different regions, causing standard self-attention mechanisms (Vaswani et al., 2017) to favor regions with larger absolute values while neglecting important relative variations. These small-scale yet dynamically critical structures are frequently averaged out or diluted at the global level, becoming what we refer to as common-mode components.

To address this limitation, our SFA branch is designed to amplify regions exhibiting strong relative variation in the spatio-temporal dimensions of the flow field. We start from the differential attention operator $\mathtt{DiffAtten}(\cdot)$ (Ye et al., 2025) with feature $X$ and projection $Q_1, Q_2, K_1, K_2 \in \mathbb{R}^{N \times d_{\text{model}}}$:

$$[Q_1; Q_2] = XW^Q, \quad [K_1; K_2] = XW^K, \quad V = XW^V$$

$$\mathtt{Attn}_i(X) = \text{softmax}(\frac{Q_i K_i^T}{\sqrt{d}}), \quad \mathtt{DiffAtten}(X) = (\mathtt{Attn}_1(X) - \lambda\,\mathtt{Attn}_2(X))V \tag{4}$$

Furthermore, we aim for $\mathtt{Attn}_1$ to focus on more localized structures, while $\mathtt{Attn}_2$ captures the broader background context. Thus, the salient flow attention, computed as the difference between the two, effectively amplifies regions with salient and strong relative variations. We can interpret $\mathtt{Attn}_2$ as a background common-mode pathway. Specifically,

$$\mu_{\mathcal{N}(j)} = \frac{1}{|\mathcal{N}(j)|} \sum_{\kappa \in \mathcal{N}(j)} K_2[\kappa], \quad \tilde{K}_2[j] = K_2[j] - \mu_{\mathcal{N}(j)}$$

$$\tilde{\mathtt{Attn}}_2[i,j] = \begin{cases} \text{softmax}\left(\frac{Q_2[i]\tilde{K}_2[j]^\top}{\sqrt{d}}\right), & j \in \mathcal{N}(i) \\ 0, & \text{otherwise} \end{cases} \tag{5}$$

$$\text{SF-Attn}(X) = (\text{Attn}_1(X) - \lambda \, \tilde{\text{Attn}}_2(X)) \, V \tag{6}$$

$\mathcal{N}(j)$ denotes the set of $\kappa$ nearest neighbors (*e.g.,* 5 as default) of patch $j$ among all $N$ patches. By suppressing attention patterns that are similar across all positions, the differential structure suppresses attention patterns that are similar across positions, thereby reducing common-mode noise and mitigating background interference. Moreover, differential mechanisms are inherently sensitive to proportional relationships rather than absolute values, which closely aligns with core physical phenomena in fluid dynamics, such as local perturbations and energy fluxes.

**Fourier Mixing (FM) Branch.** To explicitly control the frequency components, we utilize AFNO (Guibas et al., 2021) as the backbone, and add a frequency-aware weighting coefficient:

$$\mathcal{K} \cdot u^l(t,x) = \mathcal{F}^{-1} \left( \mathbf{W}_\theta^l(\boldsymbol{\xi}) \cdot \mathcal{F}[u^l](\boldsymbol{\xi}) \right) \tag{7}$$

where $\mathcal{F}$ and $\mathcal{F}^{-1}$ denote the Fourier and inverse Fourier transforms, respectively. $\boldsymbol{\xi}$ is the frequency component index. $\mathcal{F}[u^l](\boldsymbol{\xi})$ is the Fourier transform of $u^l(t,x)$, evaluated at frequency $\boldsymbol{\xi}$. $\mathbf{W}_\theta^l(\boldsymbol{\xi})$ is a complex-valued, learnable tensor that acts as a filter in the spectral domain. Since there is mode truncation to keep high-frequency components, $\mathbf{W}_\theta^l(\boldsymbol{\xi})$ can amplify or attenuate specific frequency components. We aim for $\mathbf{W}_\theta^l(\boldsymbol{\xi})$ to further enhance the representation of high-frequency features:

$$\mathbf{W}_\theta^l(\boldsymbol{\xi}) = \left( \beta_\theta^l + \alpha_\theta^l \cdot \|\boldsymbol{\xi}\|^\eta \right) \cdot \mathbf{W}_\theta^l \tag{8}$$

where $\alpha$ and $\beta$ are the learnable scaling and shifting parameters, and $\mathbf{W}_\theta^l$ is the learnable weight at layer $l$ that control the strength of the weighting. $\eta$ (initialized as 1) is a parameter that controls how the weight scales with the frequency magnitude $\|\boldsymbol{\xi}\|$; higher frequencies are assigned greater weights.

**Frequency-aware Adaptive Fusion.** Let $u_{\text{SFA}}, u_{\text{FM}} \in \mathbb{R}^{B \times H \times W \times D}$ represent the feature maps produced by the two branches. These are concatenated along the channel axis and passed through a 1D convolutional layer followed by a sigmoid activation to generate a gating map $\mathbf{G} \in \mathbb{R}^{B \times H \times W \times 1}$:

$$\mathbf{G} = \sigma \left( \text{Conv}_{1 \times 1} \left( [u_{\text{SFA}}, u_{\text{FM}}] \right) \right), \tag{9}$$

where $[\cdot, \cdot]$ denotes channel-wise concatenation. $\text{Conv}_{1 \times 1}$ is a convolutional layer that performs dimensionality reduction to a single-channel output, and $\sigma$ applies the sigmoid function element-wise.

The fused feature map $u_{\text{fused}}$ is the element-wise weighted sum of the two feature maps:

$$u_{\text{fused}} = \mathbf{G} \odot u_{\text{SFA}} + (1 - \mathbf{G}) \odot u_{\text{FM}}, \tag{10}$$

where $\odot$ indicates element-wise multiplication, the gating map $\mathbf{G}$ is broadcast along the channel dimension to align with the shape of the input feature maps.

In particular, the frequency-aware branch $u_{\text{FM}}$ provides enhanced sensitivity to high-frequency components, which are crucial for capturing fine-scale turbulent structures such as vortices and shear layers. However, such features may be spatially sparse and vary in intensity across the flow field. By incorporating a data-driven gating map $\mathbf{G}$, the model dynamically balances contributions from the spatially attentive branch $u_{\text{SFA}}$ and the frequency-aware branch $u_{\text{FM}}$, ensuring that the fused representation remains both spectrally rich and spatially coherent.

### 3.3 FREQUENCY-AWARE SURROGATE ALIGNMENT

In this section, we propose a novel regularization approach for generative models from the perspective of feature alignment. Inspired by the idea of external representation supervision introduced in REPA (Yu et al., 2024), we introduce an external pretrained encoder to guide the learning of the internal representations within the generative model. The motivation behind this strategy stems from recent findings in representation learning. Specifically, Park et al. (Park et al., 2023) observe that different pretraining paradigms emphasize different spectral components of the data: masked modeling approaches such as MAE (Shu et al., 2022) tend to capture high-frequency features, while contrastive learning frameworks like DINO (Oquab et al., 2023) are biased toward low-frequency structures. Since fluid dynamics data often exhibit rich high-frequency patterns, such as vortices, shear layers, and turbulent fluctuations, we choose to employ MAE as the foundation for constructing our external representation space.

To this end, we perform MAE pretraining on fluid simulation data. The training process begins by randomly masking a large portion (typically 75%) of spatial patches from each input sample. This

encourages the model to infer the missing regions from surrounding contextual cues, forcing the encoder to learn semantically meaningful and spatially coherent features. We use a Video Vision Transformer (ViViT) (Arnab et al., 2021) as the encoder backbone, which operates only on the visible patches. The encoder's output is then passed to a shallow transformer decoder, which reconstructs the full input field by predicting the values of the masked regions. The model is trained to minimize the MSE between the predicted and ground truth over the masked regions. Once pretrained, the MAE encoder is frozen and used as a surrogate feature extractor to guide the training of the generative model. Specifically, we enforce alignment between the intermediate representations of FourierFlow and those of the MAE encoder at selected feature layers. The total training objective of the generative model thus combines the primary flow matching loss and the alignment loss $\mathcal{L}_{\text{Total}} = \mathcal{L}_{\text{CFM}} + \gamma \cdot \mathcal{L}_{\text{Align}}$.

## 4 THEORETICAL ANALYSIS

To understand the fundamental limitations of diffusion models in learning turbulent dynamics, we formally analyze how frequency components evolve under the forward and backward diffusion processes. We show that generative models inherently exhibit a spectral bias, where high-frequency signals are corrupted earlier than low-frequency ones during the forward process. This bias results in generative models that preferentially reconstruct low-frequency structures while struggling to recover fine-scale turbulent features.

Let $\mathbf{x}_t$ follow the stochastic differential equation defined by the forward diffusion process: $d\mathbf{x}_t = g(t) d\mathbf{w}_t$, and let $\hat{\mathbf{x}}_0(\omega)$ denote the Fourier transform of the initial signal at frequency $\omega$. And let $\hat{\varepsilon}_t(\omega)$ denote the Fourier transform of the accumulated noise in the forward diffusion process up to time $t$, we have:

**Theorem 4.1 (Spectral Bias in Generative Models)** *Assume that the power spectral density of the initial signal follows a power-law decay, i.e., $|\hat{\mathbf{x}}_0(\omega)|^2 \propto |\omega|^{-\alpha}$ for some $\alpha > 0$. Then the time $t_\gamma(\omega)$ at which the signal-to-noise ratio (SNR) at frequency $\omega$ drops below a threshold $\gamma$ satisfies: $t_\gamma(\omega) \propto |\omega|^{-\alpha}$. In other words, higher-frequency components reach the noise-dominated regime earlier in the diffusion process.*

This result follows directly from the following lemmas.

**Lemma 1 (Spectral Variance of Diffusion Noise)** *For all frequencies $\omega$, the variance of $\hat{\varepsilon}_t(\omega)$ is given by: $\mathbb{E}[|\hat{\varepsilon}_t(\omega)|^2] = \int_0^t |g(s)|^2 ds$, which is constant across $\omega$.*

**Lemma 2 (Frequency-dependent Signal-to-Noise Ratio)** *Assume the initial signal $\mathbf{x}_0$ has a Fourier spectrum $\hat{\mathbf{x}}_0(\omega)$, and the diffusion follows $d\mathbf{x}_t = g(t)d\mathbf{w}_t$. Then the signal-to-noise ratio (SNR) at frequency $\omega$ is: $SNR(\omega) = \frac{|\hat{\mathbf{x}}_0(\omega)|^2}{\int_0^t |g(s)|^2 ds}$.*

**Lemma 3 (Time to Reach SNR Threshold)** *Fix a signal-to-noise ratio threshold $\gamma > 0$. Then, the time $t_\gamma(\omega)$ at which the SNR of frequency $\omega$ drops below $\gamma$ is given by: $\int_0^{t_\gamma(\omega)} |g(s)|^2 ds = \frac{|\hat{\mathbf{x}}_0(\omega)|^2}{\gamma}$.*

Together, Lemmas 1–3 demonstrate that due to the power-law decay of natural signals in the frequency domain, higher-frequency components are corrupted earlier than low-frequency components during forward diffusion. As a consequence, the generative model learns to reconstruct low-frequency information first and may fail to recover critical high-frequency features, such as vortices and shear layers, that are essential for accurate turbulence modeling. Proof is provided in Appendix H.

## 5 EXPERIMENTS

In this section, we aim to answer the following questions with solid experimental validation: **Q1 :** Which model is the best for multi-step complex fluid dynamics simulation? **Q2 :** Can the adaptive guidance of frequency-aware features encourage the generative model to overcome the spectral bias? **Q3 :** Is the salient flow attention mechanism of FourierFlow prone to reduce common-mode noise when simulating complex fluid dynamics? **Q4 :** Can the generative flow model obtain better guidance from the pre-trained surrogate model? **Q5 :** Does FourierFlow exhibit better generalization ability on out-of-distribution conditions? **Q6 :** Does FourierFlow exhibit better generalization ability on longer time steps? **Q7 :** Does FourierFlow exhibit better generalization ability on noisy input?

We address Q1 in Section 5.2. The answers to Q2, Q3, and Q4 are provided in Section 5.3, along with a comprehensive ablation study. More analysis details on spectral bias can be found in Appendix D. Q5 and Q6 are discussed in Section 5.4, and Q7 is addressed in Section E. Experiments for turbulence are

Table 1: **Results on multi-step turbulent flow modeling.** RMSE represents root mean square error, nRMSE is normalized RMSE, and Max_Err computes the maximum error of local worst case. To ensure a fair comparison, all models start with the same initial step. The surrogate model generates multi-step outputs autoregressively, while the generative model produces them directly.

| Model | Parameter | Compressible N-S (M=0.1) | | | Compressible N-S (M=1.0) | | | Shear Flow | | |
|---|---|---|---|---|---|---|---|---|---|---|
| | | MSE↓ | nRMSE↓ | Max_ERR↓ | MSE↓ | nRMSE↓ | Max_ERR↓ | MSE↓ | nRMSE↓ | Max_ERR↓ |
| *Autoregressive Surrogate models* | | | | | | | | | | |
| 2D FNO Li et al. (2020) | 12.4M | 0.1542 | 0.2965 | 3.1079 | 0.2816 | 0.4162 | 4.2707 | 0.7267 | 0.8956 | 6.6231 |
| FFNO (Tran et al., 2021) | 15.8M | 0.1014 | 0.2671 | 3.2142 | 0.2537 | 0.4115 | 4.2765 | 0.7045 | 0.8829 | 6.3019 |
| OFormer (Li et al., 2022) | 36.9M | 0.1349 | 0.2795 | 1.9742 | 0.2120 | 0.3954 | 4.0198 | 0.7022 | 0.8834 | 6.1064 |
| DPOT* (Hao et al., 2024) | 102M | 0.0628 | 0.2207 | 1.2017 | 0.1679 | 0.3276 | 3.3367 | 0.6842 | 0.7322 | 5.5641 |
| *Multi-step Surrogate models* | | | | | | | | | | |
| ViViT (Arnab et al., 2021) | 88.9M | 0.0826 | 0.2765 | 2.3558 | 0.1620 | 0.3518 | 3.4271 | 0.6294 | 0.6434 | 5.2071 |
| 3D FNO (Li et al., 2020) | 36.2M | 0.0772 | 0.3078 | 4.4769 | 0.1972 | 0.3929 | 3.6953 | 0.6991 | 0.7595 | 5.6932 |
| Ours-Surrogate | 161M | 0.0519 | 0.2033 | 1.6028 | 0.1008 | 0.3102 | 3.6852 | 0.6802 | 0.7338 | 5.5248 |
| *Next-step Generative models + Rollout* | | | | | | | | | | |
| DiT* (Peebles & Xie, 2023) | 88.2M | 0.1024 | 0.2598 | 2.2013 | 0.1749 | 0.3629 | 3.7219 | 0.6732 | 0.7842 | 5.8311 |
| DiT-DDIM* (Song et al., 2020) | 88.2M | 0.0819 | 0.2278 | 1.7908 | 0.1533 | 0.3217 | **3.2506** | 0.6699 | 0.7901 | 5.8553 |
| PDEDiff (Shysheya et al., 2024) | 53.2M | 0.0982 | 0.2452 | 1.9523 | 0.1198 | 0.3106 | 3.3087 | 0.6206 | 0.6894 | 5.2147 |
| SiT* (Ma et al., 2024) | 88.2M | 0.1004 | 0.2562 | 1.9842 | 0.1544 | 0.3652 | 3.6571 | 0.6438 | 0.7710 | 5.5826 |
| *Multi-step Generative models* | | | | | | | | | | |
| CFM* (Lipman et al., 2023) | 155M | 0.1217 | 0.3038 | 2.6365 | 0.1336 | 0.3554 | 3.3983 | 0.6224 | 0.6802 | 5.1093 |
| DYffusion (Rühling et al., 2023) | 109M | 0.0619 | 0.2098 | 1.2098 | 0.1965 | 0.3742 | 3.8762 | 0.6572 | 0.7402 | 5.2783 |
| STDiT* (Peng et al., 2025) | 169M | 0.0642 | 0.1955 | 1.1352 | 0.1125 | 0.3041 | 3.2798 | 0.5908 | 0.6412 | 5.1772 |
| **FourierFlow (ours)** | 161M | **0.0277** | **0.1530** | **0.9625** | **0.0955** | **0.2868** | 3.2551 | **0.5811** | **0.6209** | **5.0992** |

implemented on `compressible N-S` (Takamoto et al., 2022) and `shear flow` (Ohana et al., 2024) datasets, both of which involve highly nonlinear, multi-scale dynamics with sharp gradients and complex spatiotemporal structures, posing significant challenges for accurate turbulence modeling. We use 90% of the data for training. Please refer to Appendix G for more dataset details.

## 5.1 EXPERIMENTAL SETUP

**Datasets.** We evaluate on two benchmark datasets: Compressible N-S from PDEBench (Takamoto et al., 2022) and Shear Flow from Well (Ohana et al., 2024). Unless otherwise specified, each dataset is randomly split into 80% training, 10% validation, and 10% test sets, with a default spatial resolution of $128 \times 128$. The time step per generation is set to 4.

**Evaluation Metrics.** To ensure a comprehensive and objective evaluation, we employ a diverse set of quantitative metrics. First, we assess global reconstruction accuracy using the Mean Squared Error (MSE) and its normalized counterpart (nRMSE). In addition, we report the Maximum Absolute Error (MAX_ERR) to capture the worst-case deviation across the domain.

**Baselines.** We select state-of-the-art methods in video generation and neural PDE solvers as baselines for comparison. For autoregressive surrogate models, we consider (**1**) 2D Fourier Neural Operator (FNO) (Li et al., 2020), (**2**) Factorized FNO (FFNO) (Tran et al., 2021), (**3**) Operator Transformer (OFormer) (Li et al., 2022), (**4**) Denoising Operator Transformer (DPOT) (Hao et al., 2024). For multi-step surrogate models, we consider (**1**) Video Vision Transformer (ViViT) (Arnab et al., 2021), (**2**) 3D FNO (Li et al., 2020), (**3**) FourierFlow with surrogate training (Ours Surrogate). The models that use next-step generative prediction followed by rollout for multi-step forecasting include: (**1**) Diffusion Transformers (DiT*) (Peebles & Xie, 2023) (∗ means re-implementation), (**2**) Denoising Diffusion Implicit Models (DiT-DDIM*) (Song et al., 2020), (**3**) PDEDiff (Shysheya et al., 2024), (**4**) Scalable Interpolant Transformers (SiT*) (Ma et al., 2024). For multi-step generative models, we consider (**1**) Conditional Flow Matching (CFM*) (Lipman et al., 2023), (**2**) DYffusion (Rühling et al., 2023), (**3**) Spatial-temporal Diffusion Transformer (STDiT*) (Peng et al., 2025). Details can be referenced in Appendix F.

## 5.2 MAIN RESULTS

To address the question of which modeling paradigm achieves the best performance for multi-step, complex fluid dynamics simulations, we evaluate a range of state-of-the-art models on three representative turbulence scenarios, as shown in Table 1. We see that FourierFlow has achieved the sota with several key findings from the evaluation. First, our proposed FourierFlow achieves state-of-the-art performance across all scenarios, outperforming the second-best method by approximately

20% on average. This demonstrates its superior capability in modeling turbulent dynamics. Second, auto-regressive surrogate models generally underperform compared to other approaches. This may be attributed to the use of teacher forcing during training, which reduces the model's robustness to distributional shifts during long-term rollout, an issue particularly pronounced in complex fluid systems. Third, multi-step surrogate models remain highly promising. Their performance is close to that of the best generative models, and due to their higher training efficiency, surrogate approaches continue to be the dominant choice in practice. This highlights a valuable direction for future research, improving generative modeling for turbulence while maintaining the efficiency and stability of surrogate architectures. Lastly, we note that next-step generative models combined with rollout tend to perform worse than direct multi-step generation approaches. These methods also suffer from performance degradation due to error accumulation and distributional shift during inference, similar to what is observed in prediction tasks.

### 5.3 ABLATION STUDY

This section will provide important analysis of our inner components.

**Ablation on frequency-aware generation.** To answer can the adaptive guidance of frequency-aware features encourage the generative model to overcome the spectral bias, we implement extension experiments on compressible N-S with three important variants: FourierFlow **w/o FM**, FourierFlow **w/o** $\mathbf{W}_\theta^l(\boldsymbol{\xi})$ and FourierFlow **w. VF**. FourierFlow **w/o FM** refers to the variant of our method where the Fourier Mixing branch is removed, along with the adaptive fusion mechanism. This configuration eliminates the integration of high-frequency information. As shown in Figure 4, this variant exhibits a significant performance drop, highlighting the importance of frequency-aware feature fusion. FourierFlow **w/o** $\mathbf{W}_\theta^l(\boldsymbol{\xi})$ denotes the ablation variant where the learnable frequency-dependent weights used to modulate high-frequency components are removed. This comparison evaluates the importance

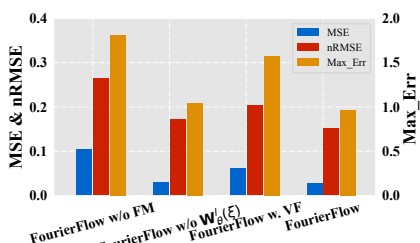

Figure 4: Ablation on frequency-aware learning on compressible N-S.

of explicitly controlling the contribution of high-frequency features. FourierFlow **w. VF** replaces our adaptive fusion mechanism with simple element-wise addition. It also leads to a notable performance degradation, underscoring the complexity of multi-frequency interactions in turbulence and the necessity of adaptive integration across different frequencies.

**Ablation on frequency-aware alignment.** To answer Q4, we conduct a systematic sensitivity analysis of the alignment loss coefficient. Specifically, we perform a grid search over the values {0, 0.001, 0.01, 0.05, 0.1, 0.5}. All experiments are conducted on compressible N-S simulations using the same settings as in the main results. As shown in Figure 5, the model achieves its best performance when the alignment coefficient is set to 0.01. Notably, deviations in either direction lead to a significant drop in performance. In particular, when the coefficient is set to 0 (*i.e.,* no alignment) or 0.5 (*i.e.,* overly emphasizing alignment), the model suffers a performance degradation of more than 20%. This ablation study not only confirms the effectiveness of alignment-based regularization, but

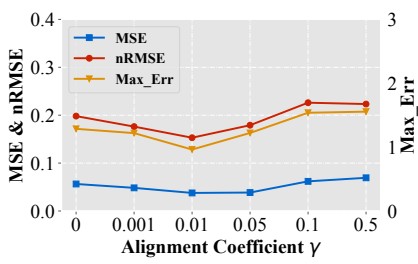

Figure 5: Ablation study on frequency-aware surrogate alignment on compressible N-S.

also shows that the model is relatively robust within the range of 0.01 to 0.05.

**Ablation on common-mode noise reduction.** To answer is the salient flow attention mechanism of FourierFlow prone to reduce common-mode noise, we introduce two key ablation variants: FourierFlow **w. SA** and FourierFlow **w/o SFA**. All of the experiments are implemented in a compressible N-S simulation and follow the same setting as in main results. The FourierFlow **w. SA** variant replaces our proposed attention mechanism with a standard self-attention module, which is widely used in current generative models. As shown in Figure 6, this modification leads to a significant performance drop, indicating that conventional attention scores may hinder the effectiveness of turbulence modeling. Further analysis of the attention distributions in Appendix C reveals that FourierFlow is capable of effectively suppressing common-mode interference, a critical factor in

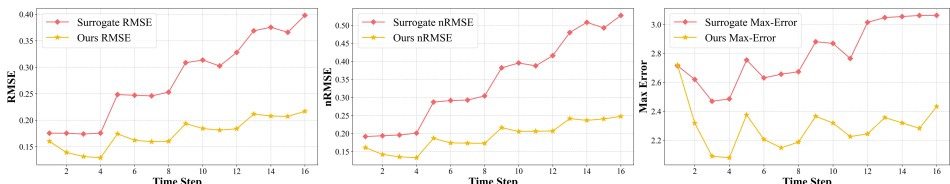

Figure 8: Comparison of multi-step generalization between the surrogate model and ours. capturing localized dynamic structures. The FourierFlow **w/o SFA** variant removes the entire SFA branch from the model. This comparison highlights the importance of the attention-based pathway in driving progressive optimization within the generative model.

## 5.4 GENERALIZATION ANALYSIS

To assess the generalization and practical applicability of FourierFlow, we conduct generalization tests under unseen conditions and longer time steps.

**Generalization analysis on out-of-distribution initial conditions.** To answer "does FourierFlow exhibit better generalization ability on out-of-distribution conditions", we evaluate zero-shot generalization performance on five compressible N-S datasets, and separately test the two cases with Mach numbers of 0.1 and 1. As shown in Figure 7, the x-axis represents different values of shear viscosity and bulk viscosity respectively, while the y-axis shows the evaluation result. The gray background indicates that the initial condition distributions of the corresponding data

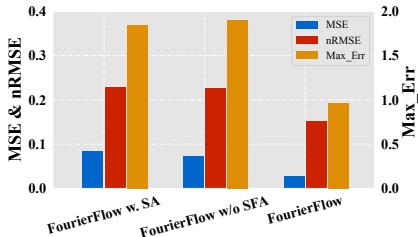

Figure 6: Ablation on common-mode noise suppression on compressible N-S.

are outside the distribution of the training data. While all models experience some degree of performance degradation under these shifted initial conditions, our proposed FourierFlow consistently demonstrates superior generalization capability compared to the SOTA surrogate baseline. These results indicate that FourierFlow is not only capable of modeling complex turbulent dynamics, but also robust to changes in real-world physical configurations.

**Generalization analysis on long-horizon rollouts.** To rigorously evaluate the temporal generalization ability, we perform long-term rollouts using the multi-step generative model trained on the compressible N-S. The surrogate baseline predicts four steps at a time and recursively rolls out using its own predictions. In contrast, our FourierFlow samples four-step trajectories conditioned on the initial state distribution and continues rollout using these sampled trajectories, effectively reducing error propagation. As illustrated in Figure 8, the benefit of this design

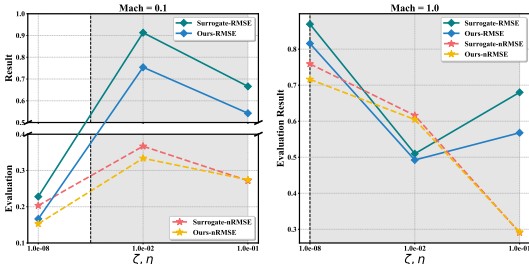

Figure 7: Generalization on compressible N-S .

becomes more pronounced at longer horizons. Under moderate flow conditions (M=0.1), our method exhibits consistently lower RMSE and nRMSE across all rollout steps, maintaining numerical stability even after hundreds of predicted steps. For the more challenging high-Mach regime (M=1.0), the performance gap widens further: the surrogate model experiences rapid error amplification and eventually diverges, whereas our method sustains physically plausible predictions, suggesting better preservation of temporal coherence and flow invariants. This demonstrates that FourierFlow not only achieves superior short-term accuracy but also provides robust generalization over extended rollouts, which is crucial for practical scientific simulation and downstream tasks.

**Generalization on noise robustness and scaling ability** can be seen in Appendix C and E.

## 6 CONCLUSION

In this paper, we have introduced FourierFlow, a frequency-aware generative framework that addresses key limitations in turbulence modeling, such as *spectral bias* and *common-mode noise*. Through a novel dual-branch architecture and surrogate feature alignment, it achieves superior accuracy, physical consistency, and generalization across complex fluid dynamics tasks.

## ETHICS STATEMENT

This work proposes a frequency-aware generative framework that addresses key limitations in turbulence modeling. All datasets are collected from public sources, without involving sensitive information.

## REPRODUCIBILITY STATEMENT

We have made significant efforts to ensure the reproducibility of FourierFlow. All datasets used in this study are collected from public benchmarks. Details are provided in Appendix G. Our codebase is publicly released in an anonymous repository https://anonymous.4open.science/r/FourierFlow-847D.

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

# APPENDIX

The appendix is organized as follows:

- We first present a comprehensive symbol glossary (Table 2) to unify notation across the paper.
- It then supplies additional empirical evidence on scaling studies in Section C, showing how FourierFlow's accuracy varies with parameter count (Figure 9)
- Section D with frequency-resolved evaluation confirms that existing diffusion models exhibit strong spectral bias, favoring low-frequency reconstruction.
- Next, we have incorporated additional robustness experiments and provided extended baseline details in Section F for completeness.
- A dedicated Section G details the datasets, including governing equations, boundary-condition setups, and turbulence-inflow formulations, complemented by multiple flow-field visualizations.
- Theoretical material in Section H follows: a frequency-domain analysis of diffusion and conditional flow-matching processes that formulates spectral bias and motivates the model's design.
- Subsequent Section I and J break down architecture components (SFA branch, Fourier mixing branch, surrogate encoder) and the training protocol, covering optimization hyper-parameters, hardware, and runtime statistics.
- The appendix concludes with discussions of limitations, future research directions, and broader societal impact in Section K. Finally, visualization analysis is provided in Section L.

## A    THE USE OF LARGE LANGUAGE MODELS (LLMS)

LLMs were not involved in the research ideation or the writing of this paper.

## B    SYMBOL

Table 2 catalogues the principal mathematical symbols used throughout FourierFlow, clarifying each definition to eliminate ambiguity across methods.

## C    MORE SUPPLEMENTARY RESULTS

**Scaling ability.** To probe the capacity limits of our generative PDE solver, we widen FourierFlow's hidden dimension from 384 to 512 and 768, increasing the parameter count from 82 M ("Small") to 161 M ("Normal") and 353 M ("Big"). As Figure 9 shows, the jump from 82 M to 161 M slashes error across all three metrics, confirming that additional width lets the model capture finer-scale dynamics. Pushing to 353 M, it still delivers modest gains. FourierFlow obeys the familiar scaling curve: performance rises steeply with capacity until the model saturates the information content of the training data, after which returns diminish. In our setting, the 161 M parameter variant offers the best accuracy–efficiency trade-off, cutting the primary error metric by about 60% relative to the 82 M model while requiring less than half the resources of the 353 M model. Going forward, we plan to (i) explore depth-wise scaling and (ii) enlarge the training corpus, two orthogonal directions that theory predicts should shift the saturation point and unlock further accuracy gains.

## D    SUPPLEMENTARY ANALYSIS ON SPECTRAL BIAS

Generative diffusion models exhibit a well-known tendency to reconstruct coarse, low-frequency structures first, while struggling to recover fine-scale, high-frequency turbulent features, a phenomenon commonly referred to as *spectral bias*. To verify and quantify this behavior, we augment baseline diffusion models with a frequency-domain RMSE loss and adopt the frequency-resolved RMSE (**fRMSE**) metric introduced in PDEBench (Takamoto et al., 2022). The fRMSE metric decomposes the solution spectrum into three disjoint frequency bands (low, mid, and high) and computes the RMSE contribution from each band, thus enabling a more fine-grained assessment of spectral fidelity.

Table 2: List of symbols used throughout the paper.

| Symbol | Meaning |
|---|---|
| $N$ | Number of patches (sequence length) |
| $d_{\text{model}}$ | Model hidden dimension |
| $W^Q,\ W^K,\ W^V$ | Query / Key / Value projection matrices |
| $Q_1,\ Q_2$ | Queries in differential attention |
| $K_1,\ K_2$ | Keys in differential attention |
| $V$ | Value projections |
| $d$ | Scaling dimension in attention ($\sqrt{d}$) |
| $\lambda$ | Balancing coefficient in differential attention |
| $\mathcal{N}(j)$ | $\kappa$ nearest neighbours of patch $j$ |
| $\kappa$ | Neighbourhood size (default 5) |
| $\mu_{\mathcal{N}(j)}$ | Mean of $K_2$ within $\mathcal{N}(j)$ |
| $\tilde{K}_2[j]$ | Mean-centred $K_2$ at patch $j$ |
| $\mathcal{K}$ | AFNO spectral operator |
| $u^l(t,x)$ | Feature field at layer $l$ and location $(t,x)$ |
| $\mathcal{F},\ \mathcal{F}^{-1}$ | Fourier / inverse-Fourier transforms |
| $\boldsymbol{\xi}$ | Spatial frequency index |
| $\mathbf{W}_\theta^l(\boldsymbol{\xi})$ | Complex spectral weight at layer $l$ |
| $\alpha_\theta^l,\ \beta_\theta^l$ | Learnable scaling / bias in FM branch |
| $\eta$ | Exponent controlling frequency weighting (init. 1) |
| $u_{\text{SFA}},\ u_{\text{FM}}$ | Feature maps from SFA / FM branches |
| $\mathbf{G}$ | Gating map produced by $1\times1$ conv + sigmoid |
| $\sigma(\cdot)$ | Sigmoid activation function |
| $\gamma$ | Alignment coefficient |
| $\odot$ | Element-wise (Hadamard) product |
| $\mathbb{E}[\cdot]$ | Expectation operator |
| $\mathcal{L}_{\text{CFM}}$ | Core flow-matching loss |
| $\mathcal{L}_{\text{Align}}$ | Representation alignment loss |
| $\mathcal{L}_{\text{Total}}$ | Total loss ($\mathcal{L}_{\text{CFM}} + \gamma\mathcal{L}_{\text{Align}}$) |

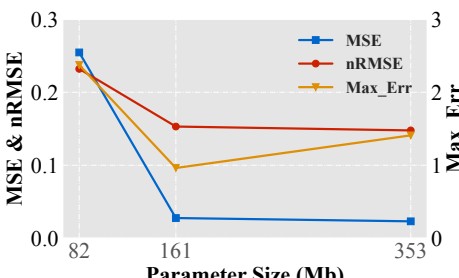

Figure 9: Scaling ability of our proposed FourierFlow.

Table 3 reports the performance of representative diffusion-based models on `compressible N-S` ($M = 0.1$). Both DiT and STDiT achieve reasonable accuracy on low-frequency components but suffer from significant degradation in the high-frequency regime. In contrast, our proposed method substantially reduces errors across all frequency bands, demonstrating a marked improvement in preserving small-scale turbulent structures.

Table 3: Frequency-resolved error analysis on Compressible Navier–Stokes ($M = 0.1$). Our method exhibits the lowest RMSE across all frequency bands, confirming its ability to mitigate spectral bias.

| Method | RMSE↓ | nRMSE↓ | fRMSE$_{\text{low}}$ ↓ | fRMSE$_{\text{mid}}$ ↓ | fRMSE$_{\text{high}}$ ↓ |
|---|---|---|---|---|---|
| DiT | 0.1024 | 0.2598 | 0.0558 | 0.0592 | 0.0948 |
| STDiT | 0.0642 | 0.1955 | 0.0447 | 0.0519 | 0.0733 |
| **Ours** | **0.0277** | **0.1530** | **0.0148** | **0.0181** | **0.0069** |

## E   SUPPLEMENTARY ROBUSTNESS RESULTS

Recent work (Chen et al., 2024) has revealed that diffusion-based generative models are robust to the noise attack. Specifically, when the input $x$ to a diffusion model is perturbed by $\delta$ (so that it becomes $x + \delta$), the resulting change in the model's output is guaranteed to be no more than $\mathcal{O}(\delta)$. To validate the robustness, we conduct an experiment in which standard Gaussian noise is injected into the training data, and subsequently compare the robustness of the surrogate model against our generative approach. As shown in Table 4, FourierFlow-$\delta$ is noticeably less affected by noise perturbations compared with Ours-Surrogate-$\delta$.

| | | Compressible N-S | | |
|---|---|---|---|---|
| MODEL | PARAMETER | RMSE↓ | NRMSE↓ | MAX_ERR↓ |
| Ours-surrogate | 161M | 0.0519 | 0.2033 | 1.6028 |
| Ours-surrogate-$\delta$ | 161M | 0.0738 | 0.2429 | 2.5319 |
| FourierFlow | 161M | 0.0277 | 0.1530 | 0.9625 |
| FourierFlow-$\delta$ | 161M | 0.0459 | 0.1894 | 1.5771 |

Table 4: Evaluation of surrogate predictive and our generative models against noise attack.

## F   BASELINE DETAILS

### F.1   FOURIER NEURAL OPERATOR (FNO)

FNO (Li et al., 2020) is a neural framework that learns mappings between infinite-dimensional function spaces. It parameterises the integral kernel in the Fourier domain and processes data through a stack of Fourier layers, each of which applies a linear transformation in frequency space, effectively performing global convolutions at $\mathcal{O}(N \log N)$ cost. In our study we examine two settings. (1) 2D FNO. The input is a single two-dimensional flow field, and the model predicts one future time step at a time, iterating forward. (2) 3D FNO. The input is a sequence of flow fields spanning multiple steps, and the model directly forecasts several future time steps in a single pass.

### F.2   FACTORIZED FOURIER NEURAL OPERATOR (FFNO)

FFNO (Tran et al., 2021) refines the original FNO architecture to push neural solvers closer to the accuracy of state-of-the-art numerical and hybrid PDE methods. It replaces the dense Fourier kernels with separable spectral layers, factorizing the multidimensional convolution into lower-rank components that preserve global receptive fields while sharply reducing parameters and memory. Deeper, carefully designed residual connections stabilize gradients, allowing the network to scale to dozens of layers without divergence. During training, F-FNO adopts a trio of heuristics: the Markov assumption, which forecasts one step and rolls out to longer horizons; Gaussian noise injection, which regularities and enhances robustness; and a cosine learning-rate decay schedule that smooths optimization.

### F.3   OPERATOR TRANSFORMER (OFORMER).

OFormer (Li et al., 2022) is an attention-centric framework that dispenses with rigid grid assumptions. OFormer marries self-attention, cross-attention and lightweight point-wise MLPs to model the relationship between arbitrary query locations and input function values, whether the samples lie on uniform meshes or irregular point clouds. A novel cross-attention decoder renders the model discretisation-invariant: given a set of context points, it predicts the output at any query coordinate without retraining. To tackle time-dependent PDEs efficiently, it embeds a latent time-marching scheme that converts the evolution into an ordinary differential equation in a learned latent space, advancing the system with a fixed step size independent of the physical resolution. The resulting Transformer architecture gracefully accommodates a variable number of input points, scales to high-dimensional domains and unifies operator approximation across disparate discretisations.

### F.4 DENOISING PRE-TRAINING OPERATOR TRANSFORMER (DPOT)

DPOT (Hao et al., 2024) introduces an auto-regressive denoising pre-training strategy that makes pre-training on PDE datasets both more stable and more efficient, while retaining strong transferability to diverse downstream tasks. Its Fourier-attention architecture is inherently flexible and scalable, allowing the model to grow smoothly for large-scale pre-training. In our work, we forgo this pre-training phase to ensure a fair comparison on the same training data. Instead, we adopt DPOT's core architecture, an efficient Transformer backbone equipped with Fourier-based attention. A learnable, nonlinear transform in frequency space enables the network to approximate kernel integral operators and to learn PDE solution maps effectively.

### F.5 VIDEO VISION TRANSFORMER (VIVIT)

ViViT (Arnab et al., 2021) encodes video by first slicing it into a grid of spatio-temporal tokens, which are then processed by Transformer layers. To tame the prohibitively long token sequences that arise, the original paper introduces several factorized variants that decouple spatial and temporal interactions. Building on this idea, we utilize ViViT's spatio-temporal attention for the baseline and apply its two-stage factorization. It computes self-attention spatially, among all tokens sharing the same timestamp, followed by self-attention temporally, among tokens occupying the same spatial location across frames. This succession of spatial-then-temporal attention preserves global context while reducing the quadratic cost to $\mathcal{O}(TS^2 + ST^2)$, making the model scalable to multi-step neural PDE solving without sacrificing expressiveness.

### F.6 OURS SURROGATE

In this baseline, we keep our full backbone, including the dual-branch design and the adaptive-fusion block, but remove the cross-attention fusion with the flow field at the initial step. Instead, the model takes the initial condition alone and predicts the target state directly, and we replace our original loss with the mean-squared-error loss commonly used by deterministic predictors. This strong variant allows us to isolate and quantify the gains our flow-based generative formulation provides over conventional prediction models in multi-step turbulence modeling.

### F.7 DIFFUSION TRANSFORMER (DIT)

DiT (Peebles & Xie, 2023) is a latent diffusion model that swaps the conventional U-Net denoiser for a plain Transformer operating on VAE. By remaining almost entirely faithful to the vanilla Vision Transformer design (layer-norm first, GELU MLP blocks, sinusoidal position encoding), DiT inherits the favorable scaling laws of vision Transformers while retaining the sample quality of state-of-the-art diffusion models. In our setting, we train a DiT variant to predict a single future flow field from an initial condition; long-horizon roll-outs are obtained by autoregressively feeding each prediction back into the model. This provides a strong, architecture-matched baseline against which to measure our method in multi-step turbulence modeling.

### F.8 DIFFUSION TRANSFORMER (DIT-DDIM)

DiT-DDIM (Song et al., 2020) preserves the transformer-based denoising network while replacing the stochastic ancestral sampler with the deterministic DDIM procedure introduced by Song *et al.* (Song et al., 2020). DDIM interprets the forward diffusion process as a non-Markovian trajectory that can be exactly reversed by solving a first-order ordinary differential equation. Concretely, given a pre-trained noise prediction model $\epsilon_\theta(\mathbf{x}_t, t)$, DDIM deterministically maps a noisy sample $\mathbf{x}_t$ at timestep $t$ to the previous timestep $t-1$ via $\mathbf{x}_{t-1} = \sqrt{\alpha_{t-1}} \left( \frac{\mathbf{x}_t - \sqrt{1-\alpha_t}\,\epsilon_\theta(\mathbf{x}_t,t)}{\sqrt{\alpha_t}} \right) + \sqrt{1 - \alpha_{t-1} - \sigma_t^2}\,\epsilon_\theta(\mathbf{x}_t, t) + \sigma_t\,\mathbf{z}$, where $\alpha_t$ is the cumulative noise schedule, $\sigma_t$ controls the residual stochasticity, and $\mathbf{z} \sim \mathcal{N}(0, \mathbf{I})$.

### F.9 CONDITIONAL DIFFUSION MODELS FOR PDE (PDEDIFF)

PDEDiff (Shysheya et al., 2024) is a baseline from fluid simulation community which also utilizes generative model. It frames PDE modeling as conditional score-based diffusion and introduces three orthogonal upgrades that jointly handle forecasting and data assimilation. It designs autoregressive

sampling, which denoises one physical step at a time and stabilizes long roll-outs compared with prior all-at-once samplers. It also uses universal amortized training, where a single conditional score network is trained over randomly sampled history lengths, allowing seamless operation across look-back windows without retraining. And a hybrid conditioning scheme is used to fuse amortized conditioning with post-training reconstruction guidance.

### F.10 Scalable Interpolant Transformers (SiT)

SiT (Ma et al., 2024) is a modular family of generative models that re-architects DiT through the lens of interpolant dynamics. Instead of fixing the forward process to a simple variance-preserving diffusion, SiT adopts an interpolant framework that smoothly transports mass between the data and latent priors along learnable paths, enabling principled ablations over four orthogonal design axes: (i) time discretization, discrete DDPM-style chains versus their continuous ODE/SDE limits; (ii) model prediction target, noise, data, or velocity; (iii) choice of interpolant,simple linear blends, variance-expanding couplings, or score-matching geodesics; and (iv) sampling rule, ranging from deterministic DDIM-like solvers to stochastic ancestral samplers. SiT combines the best design choices identified in each component.

### F.11 Spatial-temporal Diffusion Transformer (STDiT)

STDiT (Peng et al., 2025) is a spatiotemporal variant that factors attention across space and time following the ViViT design. This architecture is widely utilized in video generation area. Each flow trajectory is decomposed into $T$ frames of $H \times W$ patches, yielding a token sequence $\mathbf{z}_{t,h,w}$. In every transformer block, we first apply temporal self-attention to tokens sharing spatial indices $(h, w)$, capturing motion dynamics, and then spatial self-attention within each frame to model appearance; linear projections are reused across both phases to keep parameter growth minimal. This two-stage attention replaces DiT's purely spatial mechanism, allowing STDiT to learn joint distributions over high-resolution videos with only $\mathcal{O}(T(HW)^2)$ instead of $\mathcal{O}((THW)^2)$ pairwise interactions.

### F.12 Conditional Flow Matching (CFM)

CFM (Lipman et al., 2023) learns a vector field $\mathbf{f}_\theta(\mathbf{x}, t)$ such that integrating the ODE: $\dot{\mathbf{x}} = \mathbf{f}_\theta(\mathbf{x}, t)$ transports samples from a simple prior to the data distribution along a probability path that is itself free to be optimized. This framework unifies continuous normalizing flows and diffusion models: unlike score-based diffusion, CFM trains with a single quadratic loss and yields deterministic samplers whose complexity scales with the number of ODE solver steps needed for a given error tolerance, thereby boosting both fidelity and efficiency. We transplant the CFM objective onto a ViViT backbone, factorizing the vector-field predictor into temporal-then-spatial attention blocks to exploit the local coherence of high-resolution flow fields. The resulting CFM serves as a strong baseline for turbulence synthesis.

### F.13 Dynamics-informed Diffusion (DYffusion)

DYffusion (Rühling et al., 2023) is a dynamics-aware diffusion baseline for probabilistic spatiotemporal forecasting that targets the long-horizon stability issues typical in fluid-simulation roll-outs. Unlike vanilla Gaussian-noise diffusion, DYffusion binds the model's diffusion clock to the system's physical clock: a time-conditioned stochastic interpolator learns the forward transition $\mathbf{x}_{t+\Delta t} = \mathcal{I}_\theta(\mathbf{x}_t, \Delta t, \tau)$, while a forecaster $\mathcal{F}\phi$ inverts this dynamics-coupled path, mirroring the reverse process of standard diffusion. This coupling yields three practical advantages. (i) Multi-step consistency. Because every diffusion step coincides with a physical time step, DYffusion naturally produces stable, coherent roll-outs over hundreds of frames without adversarial trajectory stitching. (ii) Continuous-time sampling. At inference time, it integrates the learned vector field with adaptive ODE solvers, traversing arbitrary-length horizons and trading accuracy for speed on-the-fly.

## G  Datasets Details

### G.1  Compressible Navier-Stokes Equation

The compressible Navier-Stokes (CNS) equation describes a fluid flow and can be written as the continuity equation, the momentum equation, and the energy equation as follows:

$$\frac{\partial \rho}{\partial t} + \nabla \cdot (\rho \mathbf{v}) = 0,$$

$$\rho\left(\frac{\partial \mathbf{v}}{\partial t} + (\mathbf{v} \cdot \nabla)\mathbf{v}\right) = -\nabla p + \eta \nabla \mathbf{v} + \left(\zeta + \tfrac{\eta}{3}\right) \nabla(\nabla \cdot \mathbf{v}), \tag{11}$$

$$\frac{\partial}{\partial t}\left(\epsilon + \frac{\rho\, v^2}{2}\right) + \nabla \cdot \left[\left(\epsilon + p + \frac{\rho\, v^2}{2}\right)\mathbf{v} - \mathbf{v} \cdot \boldsymbol{\sigma}'\right] = 0,$$

where $\rho$ is the mass density, which represents the mass per unit volume. A higher $\rho$ means the fluid is denser, leading to larger inertia and a potentially slower response to applied forces. $\mathbf{v}$ is the velocity vector of the fluid, indicating both the speed and direction of the flow. $p$ is the gas pressure, quantifying the force exerted per unit area by the fluid's molecules. An increase in $p$ can drive the fluid to expand or accelerate. $\epsilon = \dfrac{p}{\Gamma - 1}$ is the internal energy per unit volume. It represents the energy stored in the microscopic motions and interactions of the fluid particles; higher pressure leads to greater internal energy. $\Gamma = \frac{5}{3}$ is the adiabatic index (the ratio of specific heats), typical for monatomic gases. It characterizes how the fluid's pressure changes with volume under adiabatic (no heat exchange) processes. $\boldsymbol{\sigma}'$ is the viscous stress tensor, which describes the internal friction within the fluid resulting from viscosity, dissipating kinetic energy into heat. $\eta$ and $\zeta$ are the shear and bulk viscosities, respectively. The shear viscosity $\eta$ measures the fluid's resistance to deformations that change its shape (shearing), while the bulk viscosity $\zeta$ quantifies its resistance to changes in volume (compression or expansion). $\Lambda$ and $\psi$ represent additional terms, such as external potentials or other source terms. Their exact physical role depends on the context, for example, modeling capillarity, external forces, or extra diffusion effects.

As for the CNS dataset setting, PDEBench (Takamoto et al., 2022) is followed as the standard. It uses $N_d$ as the number of spatial dimensions, and the Mach number is defined by

$$M = \frac{|v|}{c_s}, \quad c_s = \sqrt{\frac{\Gamma\, p}{\rho}}, \tag{12}$$

where $c_s$ is the sound speed. For outflow boundary conditions, a common approach is to copy the state from the nearest cell to the boundary. For inflow, different strategies are used in 1D, 2D, and 3D to specify velocity, density, and pressure.

A turbulent inflow condition can be imposed by superimposing fluctuations on a base flow with nearly uniform density and pressure. The velocity component may be represented by a Fourier-type sum (analogous to Eq. (8) in certain references):

$$v_x(t, z) = \sum_{i=1}^{n} A_i \sin(k_i\, x + \phi_i), \tag{13}$$

where $n = 4$ and $A_i = \bar{v}_i/k_i$ (the exact values depend on the desired amplitude and wavenumber). A typical Mach number might be $M = 0.12$ to reduce compressibility effects. This kind of velocity field can be viewed as a simplified Helmholtz decomposition in Fourier space.

The shock-tube initial field is composed as:

$$Q_{SL}(x, t = 0) = (Q_L,\, Q_R), \tag{14}$$

where $Q = (\rho,\, p,\, \mathbf{v},\, \phi)$ denotes the fluid state. This is known as the "Riemann problem," which typically includes shock waves and rarefaction waves whose precise form depends on the Mach number and flow conditions. It often requires a numerical solver to capture the full wave structure. Such a scenario can be used to test whether a ML model truly understands the set of compressible flow equations. In practice, a second-order accurate HLLC flux-splitting scheme with a MUSCL approach is commonly employed for the inviscid part, while additional terms (*e.g.*, viscosity or other source terms) may be discretized depending on the specific requirements.

### G.2 INCOMPRESSIBLE NAVIER-STOKES EQUATION

We consider a 2D periodic incompressible shear flow, a classical and practically significant configuration in fluid dynamics. A shear flow refers to a velocity field in which layers of fluid move parallel to each other at different speeds, causing continuous deformation due to velocity gradients. Such flows appear frequently in both natural environments (*e.g.*, ocean currents, atmospheric jet streams) and engineering systems (*e.g.*, lubrication, pipeline flows).

Following the data in (Ohana et al., 2024), the fluid dynamics are governed by the incompressible N-S equations, defined over a two-dimensional periodic domain $\Omega = [0, 1]^2$:

$$\frac{\partial \mathbf{u}}{\partial t} + (\mathbf{u} \cdot \nabla)\mathbf{u} + \nabla p = \nu \Delta \mathbf{u} + \mathbf{f}, \quad \nabla \cdot \mathbf{u} = 0 \tag{15}$$

where $\mathbf{u} = (u_x, u_y)$ denotes the velocity field, $p$ is the pressure field, $\nu$ is the kinematic viscosity, $\mathbf{f}$ is an external forcing term, $\Delta$ is the Laplacian operator, and the second equation enforces the incompressibility constraint. To generate a sustained shear structure, we impose a time-independent, spatially varying forcing function of the form:

$$\mathbf{f}(x, y) = (\alpha \sin(2\pi y)\, 0), \tag{16}$$

where $\alpha$ is a constant controlling the strength of the external force in the $x$-direction, promoting shear across horizontal layers. We apply periodic boundary conditions in both $x$ and $y$ directions:

$$\mathbf{u}(0, y, t) = \mathbf{u}(1, y, t), \quad \mathbf{u}(x, 0, t) = \mathbf{u}(x, 1, t), \quad \forall t \geq 0, \tag{17}$$

ensuring that the domain forms a continuous torus and mimics unbounded behavior in a bounded computational setting.

The initial condition $\mathbf{u}_0(x, y)$ is typically a divergence-free random field or analytically constructed to align with shear characteristics (*e.g.*, $u_x \propto \sin(2\pi y)$, $u_y = 0$), allowing the system to evolve under nonlinear convection and diffusion effects.

## H THEORETICAL ANALYSIS

In the diffusion model, data is gradually perturbed through a forward diffusion process with additive noise, eventually becoming pure noise signals. In the reverse process (denoising), the model learns to reconstruct the original data from noise. Understanding how different frequency components evolve during forward diffusion is crucial for model design and performance. This paper rigorously demonstrates that at the beginning of diffusion, high-frequency information is disrupted first. As time progresses, low-frequency information is gradually affected.

Consider a continuous-time diffusion process, with initial data $\mathbf{x}_0$. As time $t \in [0, T]$ progresses, the data evolves into $\mathbf{x}_t$. This diffusion process can be described by the following stochastic differential equation:

$$d\mathbf{x}_t = f(\mathbf{x}_t, t)dt + g(t)d\mathbf{w}_t \tag{18}$$

where $\mathbf{w}_t$ is a Wiener process (Brownian motion), representing random noise. $f(\mathbf{x}_t, t)$ is the drift term. For simplification, we can set $f(\mathbf{x}_t, t) = 0$. $g(t)$ is a time-dependent diffusion coefficient that controls the strength of the noise. With this simplification, the diffusion process becomes:

$$d\mathbf{x}_t = g(t)d\mathbf{w}_t \tag{19}$$

The initial condition is $\mathbf{x}_0$.

To analyze how the diffusion process affects different frequency components, we transform the signal from the time domain to the frequency domain. Let the Fourier transform of the signal $\mathbf{x}_t$ be denoted by $\hat{\mathbf{x}}_t(\omega)$, where $\omega$ represents frequency. The linearity of the Fourier transform and the properties of the Wiener process make analysis in the frequency domain more tractable. Since the noise $d\mathbf{w}_t$ is white noise, its power spectral density is constant in the frequency domain.

## H.1 PROOF OF LEMMA 1 (SPECTRAL VARIANCE OF DIFFUSION NOISE)

**Lemma 1 (Spectral Variance of Diffusion Noise)** *For all frequencies $\omega$, the variance of $\hat{\varepsilon}_t(\omega)$ is given by*

$$\mathbb{E}\big[|\hat{\varepsilon}_t(\omega)|^2\big] = \int_0^t |g(s)|^2 \, \mathrm{d}s,$$

*which is constant in $\omega$.*

**Proof H.1** *We start from the integral form of the forward SDE, $\mathbf{x}_t = \mathbf{x}_0 + \int_0^t g(s) \, \mathrm{d}\mathbf{w}_s$, where $\mathbf{w}_s$ is a standard Wiener process (white noise). Denote $\varepsilon_t := \int_0^t g(s) \, \mathrm{d}\mathbf{w}_s$, so that in the frequency domain*

$$\hat{\mathbf{x}}_t(\omega) = \hat{\mathbf{x}}_0(\omega) + \hat{\varepsilon}_t(\omega), \tag{20}$$

*with $\hat{\varepsilon}_t(\omega) = \mathcal{F}\big[\varepsilon_t\big](\omega)$.*

**Zero mean.** *Since the Itô integral has mean zero,*

$$\mathbb{E}\big[\varepsilon_t(x)\big] = 0 \quad \implies \quad \mathbb{E}\big[\hat{\varepsilon}_t(\omega)\big] = 0. \tag{21}$$

**Second moment via Itô isometry.** *By the Itô isometry in the spatial domain,*

$$\mathbb{E}\big\|\varepsilon_t\big\|_{L_x^2}^2 = \mathbb{E} \int \Big|\int_0^t g(s) \, \mathrm{d}\mathbf{w}_s(x)\Big|^2 \, \mathrm{d}x = \int_0^t |g(s)|^2 \, \mathrm{d}s. \tag{22}$$

*Moreover, the Fourier transform $\mathcal{F}$ is an isometry on $L_x^2$, so*

$$\mathbb{E}\big\|\hat{\varepsilon}_t\big\|_{L_\omega^2}^2 = \mathbb{E}\big\|\varepsilon_t\big\|_{L_x^2}^2 = \int_0^t |g(s)|^2 \, \mathrm{d}s. \tag{23}$$

**Frequency-wise variance.** *White noise has flat spectral density and yields uncorrelated Fourier modes. Concretely, for each fixed $\omega$,*

$$\mathbb{E}\big[|\hat{\varepsilon}_t(\omega)|^2\big] \;=\; \int_0^t |g(s)|^2 \, \mathrm{d}s, \tag{24}$$

*since no additional $\omega$-dependence arises, both the Itô isometry and the $L^2$-isometry of the Fourier transform preserve the accumulated variance uniformly across frequencies. Thus the spectral variance of the diffusion noise is identical for all $\omega$.*

## H.2 PROOF OF LEMMA 2 (FREQUENCY-DEPENDENT SNR)

**Lemma 2 (Frequency-dependent Signal-to-Noise Ratio)** *Assume the initial signal $\mathbf{x}_0$ has a Fourier spectrum $\hat{\mathbf{x}}_0(\omega)$, and the diffusion follows $d\mathbf{x}_t = g(t)d\mathbf{w}_t$. Then the signal-to-noise ratio (SNR) at frequency $\omega$ is:*

$$SNR(\omega) = \frac{|\hat{\mathbf{x}}_0(\omega)|^2}{\int_0^t |g(s)|^2 ds} \tag{25}$$

**Proof H.2** *We decompose the Fourier-domain observation at time $t$ as $\hat{\mathbf{x}}_t(\omega) = \hat{\mathbf{x}}_0(\omega) + \hat{\varepsilon}_t(\omega)$, where $\hat{\varepsilon}_t(\omega) = \mathcal{F}\big[\int_0^t g(s) \, \mathrm{d}\mathbf{w}_s\big](\omega)$ is the Fourier transform of the accumulated diffusion noise.*

**Power spectral density.** *By definition, the power spectral density of $\mathbf{x}_t$ at frequency $\omega$ is*

$$S_{\mathbf{x}_t}(\omega) = \mathbb{E}\big[|\hat{\mathbf{x}}_t(\omega)|^2\big]. \tag{26}$$

*Substituting the decomposition gives*

$$\begin{aligned}
S_{\mathbf{x}_t}(\omega) &= \mathbb{E}\Big[\big|\hat{\mathbf{x}}_0(\omega) + \hat{\varepsilon}_t(\omega)\big|^2\Big] \\
&= \big|\hat{\mathbf{x}}_0(\omega)\big|^2 + 2 \, \mathrm{Re}\Big(\hat{\mathbf{x}}_0(\omega) \, \mathbb{E}\big[\hat{\varepsilon}_t^*(\omega)\big]\Big) + \mathbb{E}\big[|\hat{\varepsilon}_t(\omega)|^2\big].
\end{aligned} \tag{27}$$

*Since the Itô integral has zero mean $\mathbb{E}\big[\hat{\varepsilon}_t(\omega)\big] = 0$, the cross term vanishes and we obtain*

$$S_{\mathbf{x}_t}(\omega) = \big|\hat{\mathbf{x}}_0(\omega)\big|^2 + \mathbb{E}\big[|\hat{\varepsilon}_t(\omega)|^2\big]. \tag{28}$$

**Noise variance via Itô isometry.** *By the Itô isometry and the $L^2$-isometry of the Fourier transform,*

$$\mathbb{E}\big[|\hat{\varepsilon}_t(\omega)|^2\big] = \int_0^t |g(s)|^2 \, \mathrm{d}s, \tag{29}$$

*which is independent of $\omega$.*

**Definition of SNR.** *We define the SNR at frequency $\omega$ as the ratio of signal-power to noise-power:*

$$\mathrm{SNR}(\omega) = \frac{\big|\hat{\mathbf{x}}_0(\omega)\big|^2}{\displaystyle\int_0^t |g(s)|^2 \, \mathrm{d}s}. \tag{30}$$

### H.3 PROOF OF LEMMA 3 (TIME TO REACH SNR THRESHOLD)

**Lemma 3 (Time to Reach SNR Threshold)** *Fix a signal-to-noise ratio threshold $\gamma > 0$. Then, the time $t_\gamma(\omega)$ at which the SNR of frequency $\omega$ drops below $\gamma$ is given by:*

$$\int_0^{t_\gamma(\omega)} |g(s)|^2 ds = \frac{|\hat{\mathbf{x}}_0(\omega)|^2}{\gamma}$$

**Proof H.3** *By Lemma 2, $\mathrm{SNR}_t(\varepsilon) \leq \gamma$ if and only if*

$$\frac{|\widehat{x}_0(\varepsilon)|^2}{\int_0^t |g(s)|^2 \, ds} \leq \gamma, \tag{31}$$

*i.e.*

$$\int_0^t |g(s)|^2 \, ds \geq \frac{|\widehat{x}_0(\varepsilon)|^2}{\gamma}. \tag{32}$$

*Since the left-hand side is continuous and strictly increasing in $t$, there is a unique crossing time given by the stated integral equation.*

### H.4 PROOF OF THEOREM 1 (SPECTRAL BIAS IN GENERATIVE MODELS)

**Theorem H.1 (Spectral Bias in Generative Models)** *Assume the initial signal power obeys a power–law decay,*

$$|\widehat{x}_0(\varepsilon)|^2 \propto |\varepsilon|^{-\omega}, \quad \omega > 0.$$

*Then the threshold time $t_\gamma(\varepsilon)$ satisfies*

$$t_\gamma(\varepsilon) \propto |\varepsilon|^{-\omega},$$

*i.e., higher-frequency components (larger $|\varepsilon|$) reach the noise-dominated regime earlier.*

**Proof H.4** *Combining Lemma 3 with the power–law assumption gives*

$$\int_0^{t_\gamma(\varepsilon)} |g(s)|^2 \, ds = \frac{|\widehat{x}_0(\varepsilon)|^2}{\gamma} \propto \frac{|\varepsilon|^{-\omega}}{\gamma}. \tag{33}$$

*Since $\int_0^t |g(s)|^2 \, ds$ is monotonically increasing in $t$, we invert this relation to obtain*

$$t_\gamma(\varepsilon) \propto |\varepsilon|^{-\omega}. \tag{34}$$

*Thus as $|\varepsilon|$ grows, $t_\gamma(\varepsilon)$ decreases, proving that high-frequency modes are corrupted earlier.*

## I ARCHITECTURE DETAILS

### I.1 FLOW MATCHING TRAINING MECHANISM

To probe the aptitude of modern generators for turbulent–flow synthesis, we cast both flow and diffusion models within the *stochastic interpolant* framework of (Albergo et al., 2023). The remainder of this section outlines our learning procedure from that standpoint.

**Continuous–time formulation.** Whereas DDPMs (Ho et al., 2020) rely on a discrete, noise-perturbation chain, flow-based generators (Esser et al., 2024; Lipman et al., 2023; Liu et al., 2023) view synthesis as a continuous path $\mathbf{x}(t)$, $t \in [0,1]$, that starts from data $\mathbf{x}(0) \sim p(\mathbf{x})$ and ends in white noise $\epsilon \sim \mathcal{N}(\mathbf{0}, \mathbf{I})$:

$$\mathbf{x}(t) = \alpha_t\,\mathbf{x}(0) + \sigma_t\,\epsilon, \quad \alpha_0 = \sigma_1 = 1,\; \alpha_1 = \sigma_0 = 0, \tag{35}$$

with $\alpha_t$ decreasing and $\sigma_t$ increasing in $t$. Equation 35 induces the *probability-flow ODE*

$$\dot{\mathbf{x}}(t) = \mathbf{v}(\mathbf{x}(t), t), \tag{36}$$

whose solution at each instant matches the marginal $p_t(\mathbf{x})$.

**Velocity objective.** The drift in Equation 36 splits into two conditional expectations,

$$\mathbf{v}(\mathbf{x}(t), t) = \dot{\alpha}_t\,\mathbb{E}[\mathbf{x}(0)\,|\,\mathbf{x}(t) = \mathbf{x}] + \dot{\sigma}_t\,\mathbb{E}[\epsilon\,|\,\mathbf{x}(t) = \mathbf{x}], \tag{37}$$

which we approximate with a neural predictor $\mathbf{v}_\theta$. Matching $\mathbf{v}_\theta$ to the target $\mathbf{v}^* = \dot{\alpha}_t\,\mathbf{x}(0) + \dot{\sigma}_t\,\epsilon$ yields the MSE loss as follows:

$$\mathcal{L}_{\text{flow}} = \mathbb{E}_{t \sim \mathcal{U}(0,1)}\big[\|\mathbf{v}_\theta(\mathbf{x}(t), t) - \mathbf{v}^*(\mathbf{x}(t), t)\|^2\big]. \tag{38}$$

**Score objective and duality.** The same dynamics admit the reverse SDE:

$$d\mathbf{x}(t) = \mathbf{v}(\mathbf{x}(t), t)\,dt - \tfrac{1}{2}\,w_t\,\mathbf{s}(\mathbf{x}(t), t)\,dt + \sqrt{w_t}\,d\bar{\mathbf{w}}_t, \tag{39}$$

whose score is

$$\mathbf{s}(\mathbf{x}(t), t) = -\frac{1}{\sigma_t}\,\mathbb{E}[\epsilon\,|\,\mathbf{x}(t) = \mathbf{x}]. \tag{40}$$

A network $\mathbf{s}_\theta$ is trained with

$$\mathcal{L}_{\text{score}} = \mathbb{E}_{\mathbf{x}(0), \epsilon, t}\big[\|\sigma_t\,\mathbf{s}_\theta(\mathbf{x}(t), t) + \epsilon\|^2\big]. \tag{41}$$

Because $\mathbf{s}$ and $\mathbf{v}$ are analytically related via

$$\mathbf{s}(\mathbf{x}, t) = \frac{\alpha_t\,\mathbf{v}(\mathbf{x}, t) - \dot{\alpha}_t\,\mathbf{x}}{\sigma_t\,\dot{\alpha}_t - \alpha_t\,\dot{\sigma}_t}, \tag{42}$$

estimating either one is sufficient.

**Choosing the interpolant.** Recent work (Albergo et al., 2023) shows that any differentiable pair $(\alpha_t, \sigma_t)$ satisfying $\alpha_t^2 + \sigma_t^2 > 0$ and the boundary conditions in Equation 35 creates an unbiased bridge between data and noise. Common examples are the linear schedule $(\alpha_t, \sigma_t) = (1 - t, t)$ and the variance-preserving choice $(\alpha_t, \sigma_t) = (\cos\frac{\pi}{2}t,\ \sin\frac{\pi}{2}t)$ (Ma et al., 2024). Crucially, the diffusion coefficient $w_t$ in Equation 39 is *decoupled* from training and can be selected *post-hoc* during sampling.

**Relation to DDPMs.** Classical score-based models (*e.g.*, DDPM) can be recast as discretised SDEs whose forward pass drifts towards an isotropic Gaussian as $t \to \infty$. After training on a finite horizon $T$ (typically $10^3$ steps), generation proceeds by integrating the corresponding reverse SDE from $\mathcal{N}(\mathbf{0}, \mathbf{I})$. In that setting, $(\alpha_t, \sigma_t)$ and $w_t$ are locked in by the forward schedule, which may overly constrain design flexibility (Song & Ermon, 2019).

## I.2 Salient Flow Attention (SFA) Branch

This branch is built upon the Transformer architecture. Beyond the attention mechanism described in the main text, we now detail the remaining components. First, we apply the patch embedding technique from ViT (Dosovitskiy et al., 2020) independently to each video frame. Specifically, when extracting non-overlapping image patches of size $h \times w$ from a frame of spatial dimension $H \times W$, the number of patches per frame along height and width becomes $n_h = H/h$ and $n_w = W/w$, respectively, while $n_f = F$ corresponds to the total frame count. The second approach extends ViT's patch embedding into the temporal domain. Here, we extract spatio-temporal "tubes" of size $h \times w \times s$ with temporal stride $s$, yielding $n_f = F/s$ tube-tokens per clip. This inherently fuses spatial and temporal information at the embedding stage. When using this tube-based embedding,

we further employ a 3D transposed convolution for temporal upsampling in the decoder, following the standard linear projection and reshaping operations. To enable temporal awareness, we inject positional embeddings into the model via two strategies: (1) absolute positional encoding, which uses sine and cosine functions at different frequencies to encode each frame's absolute position in the sequence (Vaswani et al., 2017); and (2) relative positional encoding, which leverages rotary positional embeddings (RoPE) to model pairwise temporal relationships between frames (Su et al., 2024). The spatial-temporal Transformer block focuses on decomposing the multi-head attention in the Transformer block. We initially compute SFA only on the spatial dimension, followed by the temporal dimension. As a result, each Transformer block captures both spatial and temporal information. The inputs for spatial SFA and temporal SFA are $z_s \in \mathbb{R}^{n_f \times s \times d}$ and $z_t \in \mathbb{R}^{s \times n_f \times d}$, respectively.

### I.3 FOURIER MIXING BRANCH

Our FM branch is based on AFNO (Guibas et al., 2021), which overcomes the scalability and rigidity of the standard FNO. AFNO introduces three key enhancements. First, it replaces dense $d \times d$ token-wise weight matrices with a block-diagonal decomposition into $k$ blocks of size $\frac{d}{k} \times \frac{d}{k}$, enabling independent, parallelizable projections analogous to multi-head attention. Second, it replaces static, token-wise weights with a shared two-layer perceptron, $\text{MLP}(z_{m,n}) = W_2\,\sigma(W_1 z_{m,n}) + b$, so that all tokens adaptively recalibrate channel mixing using a fixed, sample-agnostic parameter set. Finally, to exploit the inherent sparsity in the Fourier domain, it applies channel-wise soft-thresholding via

$$\tilde{z}_{m,n} = S_\lambda\big(W_{m,n} z_{m,n}\big), \quad S_\lambda(x) = \text{sign}(x)\max\{|x| - \lambda, 0\}, \tag{43}$$

which simultaneously prunes low-energy modes and regularizes the network. Collectively, these modifications yield the AFNO mixer, whose block structure, weight sharing, and shrinkage jointly improve parameter efficiency, expressivity, and robustness compared to FNO and self-attention.

### I.4 PRE-TRAINED SURROGATE ARCHITECTURE

We utilize the backbone of ViViT for MAE pretraining. It extends ViT (Dosovitskiy et al., 2020) to video by tokenizing spatio-temporal inputs and applying transformer-based attention across both space and time. First, a video clip $\mathbf{V} \in \mathbb{R}^{T \times H \times W \times C}$ is mapped to tokens either by (i) uniformly sampling $n_t$ frames and extracting $n_h \times n_w$ 2D patches per frame, or (ii) extracting non-overlapping 3D "tubelets" of size $t \times h \times w$, yielding $n_t = \lfloor T/t \rfloor$, $n_h = \lfloor H/h \rfloor$, and $n_w = \lfloor W/w \rfloor$ tokens per clip. Positional embeddings are then added before feeding the sequence into a transformer encoder.

ViViT offers four architectural variants to balance modeling power and cost: *Joint Space-Time Attention*. It computes full self-attention over all $n_t n_h n_w$ tokens and captures long-range interactions from the first layer. *Factorized Encoder*. It applies a spatial transformer to each frame independently and then a temporal transformer over the resulting frame-level embeddings, reducing FLOPs from $\mathcal{O}((n_t n_h n_w)^2)$ to $\mathcal{O}((n_h n_w)^2 + n_t^2)$. *Factorized Self-Attention*. It alternates spatial and temporal self-attention within each layer, achieving the same complexity as the factorized encoder while maintaining a unified block structure. *Factorized Dot-Product Attention*. It allocates half of the attention heads to spatial neighborhoods and half to temporal neighborhoods, preserving parameter count but reducing compute to $\mathcal{O}((n_h n_w)^2 + n_t^2)$. These designs equip ViViT with flexible trade-offs between expressivity and efficiency for high-fidelity spatial-temporal understanding.

## J  IMPLEMENTATION DETAILS.

Following the original SiT implementation (Albergo et al., 2023), we integrate our proposed architectural enhancements and train the model using the AdamW optimizer with a cyclic learning-rate schedule (initial learning rate $1 \times 10^{-4}$) to facilitate periodic warm-up and cooldown phases (`torch.optim.lr_scheduler.CyclicLR`). Training is accelerated across multiple GPUs via the `Accelerator` framework, and gradients are clipped to stabilize convergence.

All experiments were conducted on $8 \times$ NVIDIA H800 80GB GPUs, achieving a sustained throughput of approximately 1.23 steps/s with a global batch size of 360. This performance can be further improved through additional engineering optimizations, such as precomputing features from the pretrained encoder.

## K    LIMITATIONS AND BROADER IMPACT

**Limitations.**   While our flow-based modeling framework demonstrates strong accuracy, it incurs substantial training costs due to the iterative sampling and high-dimensional feature mixing. Accelerating convergence remains an open challenge: potential directions include deriving mathematically principled guidance terms to reduce sampling iterations, designing more efficient transformer variants tailored to Fourier-domain operations, and developing lightweight context-encoding modules that minimize redundant computation. Addressing these aspects will be crucial for scaling our approach to even larger spatio-temporal domains and real-time applications.

**Broader Impact.**   By advancing the fidelity and scalability of turbulence simulation, our work paves the way for real-world deployment in domains such as aerodynamics, structural materials design, and plasma physics. More generally, our method for simulating complex physical systems holds promise for climate forecasting, fusion-plasma modeling, and other scientific or engineering applications. At the same time, we acknowledge potential risks: highly realistic simulations could be misused to design unsafe or unethical systems. We therefore advocate for responsible dissemination and continued vigilance to ensure that the benefits of our research are realized without adverse societal consequences.

## L    VISUALIZATION ANALYSIS

**Case study.**  We compare Ours-Surrogate and FourierFlow on the CNS benchmark by training each with four input time steps to predict the next step, and then rolling out their forecasts for 20 additional steps. Figure 10, 11 12 and 13 visualize the resulting density, pressure, and velocity fields. Both approaches reproduce the global structure of the flow, but a closer inspection reveals marked differences: the surrogate's predictions blur or miss fine-scale turbulent features, whereas our method more faithfully reconstructs the rapidly evolving details. Moreover, the surrogate often exhibits patch artifacts, a by-product of its patch-wise training regime, while our holistic, end-to-end generative optimization largely avoids this issue, though a small amount of residual noise can still be seen in a few frames.

**Empirical evidence of spectral bias.** In order to quantify and visualize the distribution of spatial frequencies in the ground-truth fields and the predicted error, we proceed as follows. For each sample index $i$ and channel $j$, we first compute the two-dimensional discrete Fourier transform $\hat{G}_{i,j}(u,v) = \text{FFT}\big(G_{i,j}(x,y)\big)$, and apply a Fourier shift so that the zero frequency component is centered. The spectral energy at each frequency bin is then given by $E_{i,j}(u,v) = \big|\hat{G}_{i,j}(u,v)\big|^2$. Next, we convert the Cartesian frequency coordinates (u,v) into a radial wavenumber $k = \sqrt{u^2 + v^2}$, and discretize $k$ into integer bins $k = 0, 1, 2, \ldots, K_{\max}$. For each integer bin $k$, we accumulate the total spectral energy $E_{i,j}(k) = \sum_{(u,v)\,:\,\lfloor\sqrt{u^2+v^2}\rfloor=k} E_{i,j}(u,v)$, and plot $\log E_{i,j}(k)$ against $k$ as a bar chart. Each subplot in Figure 14 corresponds to one channel $j$ of sample $i$, with the horizontal axis denoting the wavenumber $k$ and the vertical axis denoting the logarithm of the integrated spectral energy. This presentation makes it easy to compare the decay of energy across scales, and to assess how well different models preserve or distort information at each spatial frequency.

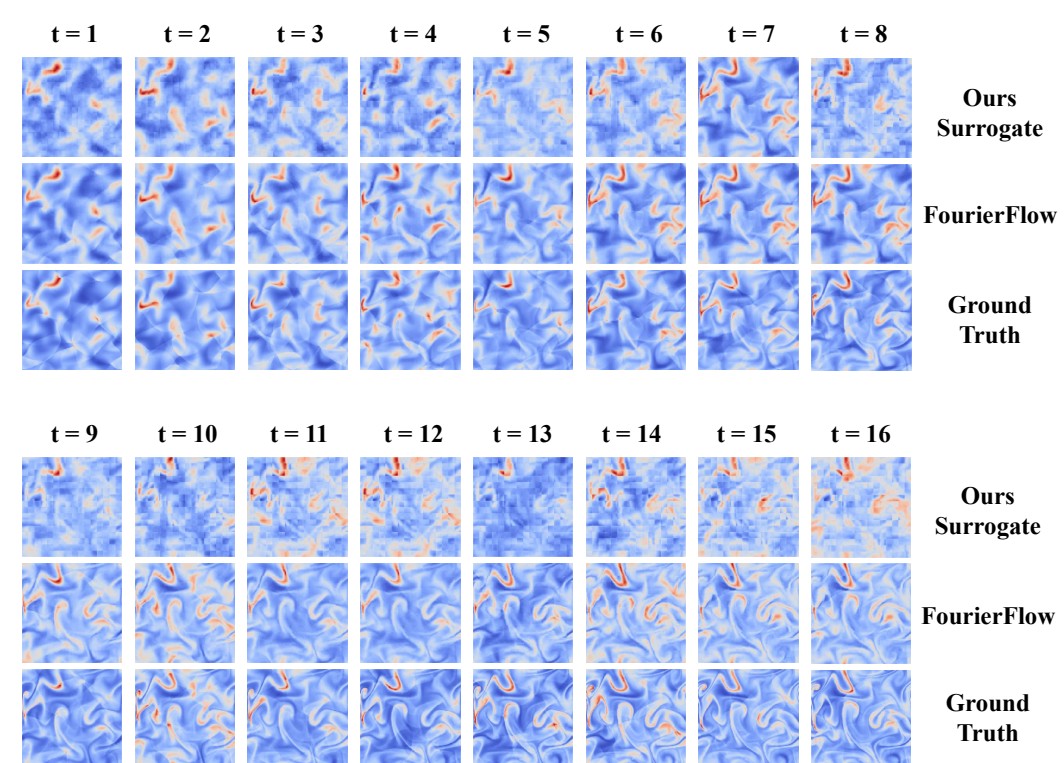

Figure 10: Visualization of the density fields produced by two methods for the compressible N-S.

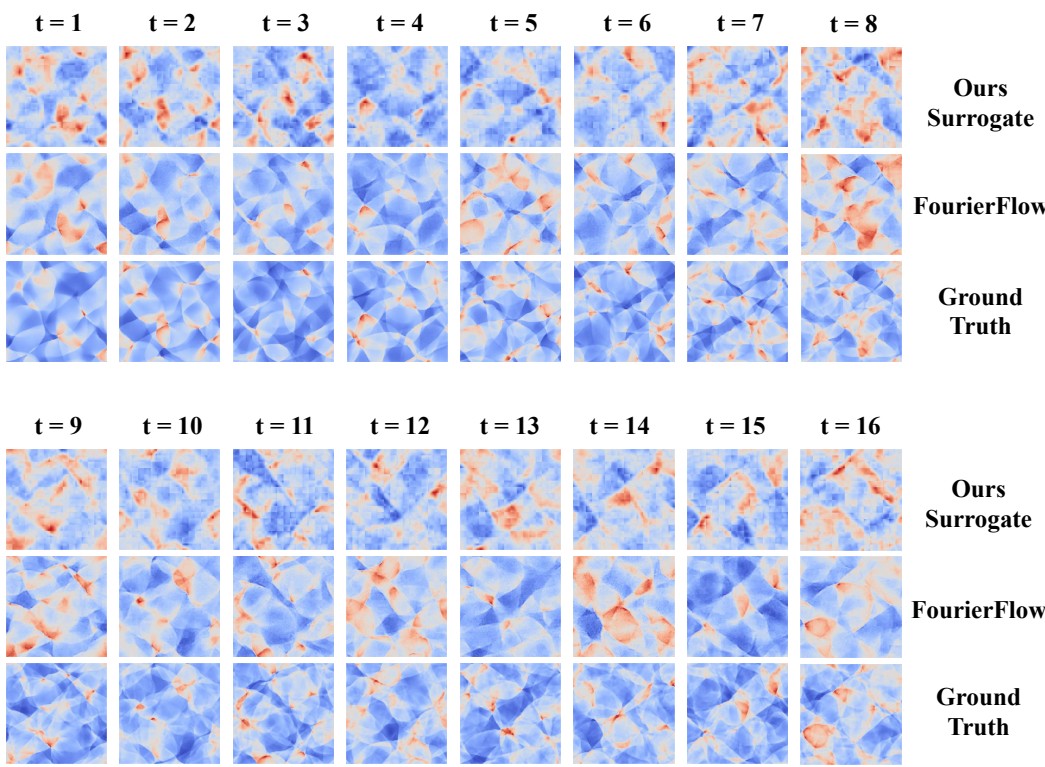

Figure 11: Visualization of the pressure fields produced by two methods for the compressible N-S.

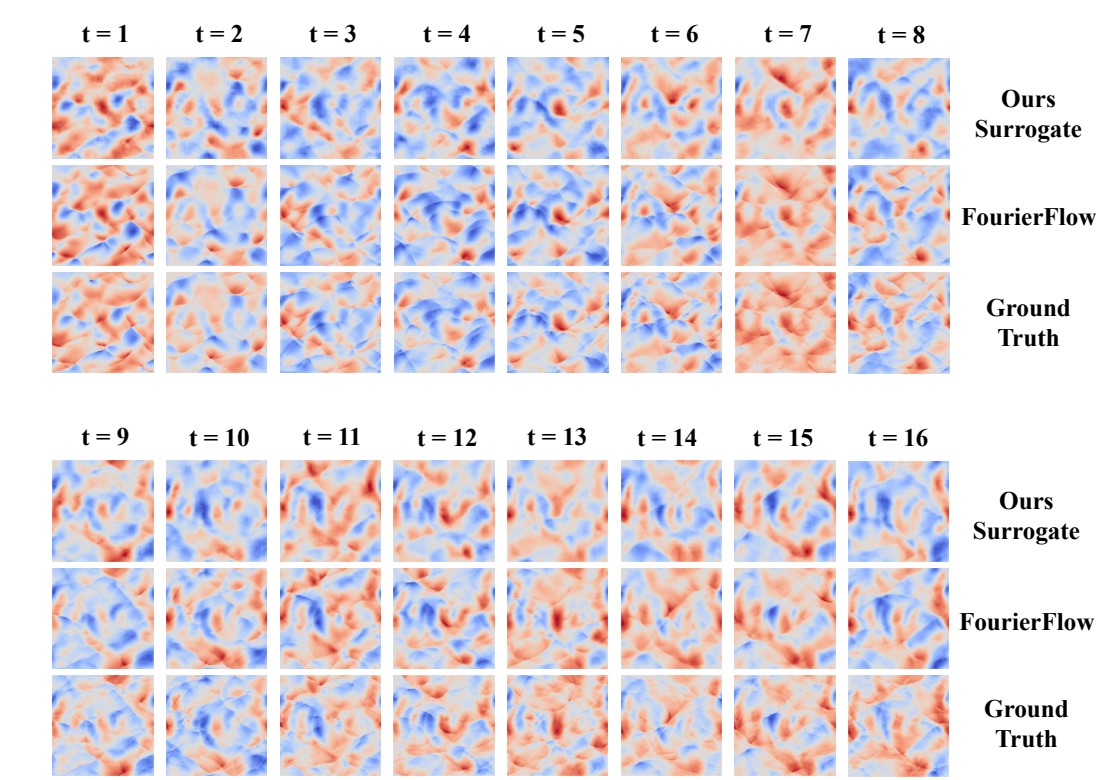

Figure 12: Visualization of the velocity-x fields produced by two methods for the compressible N-S.

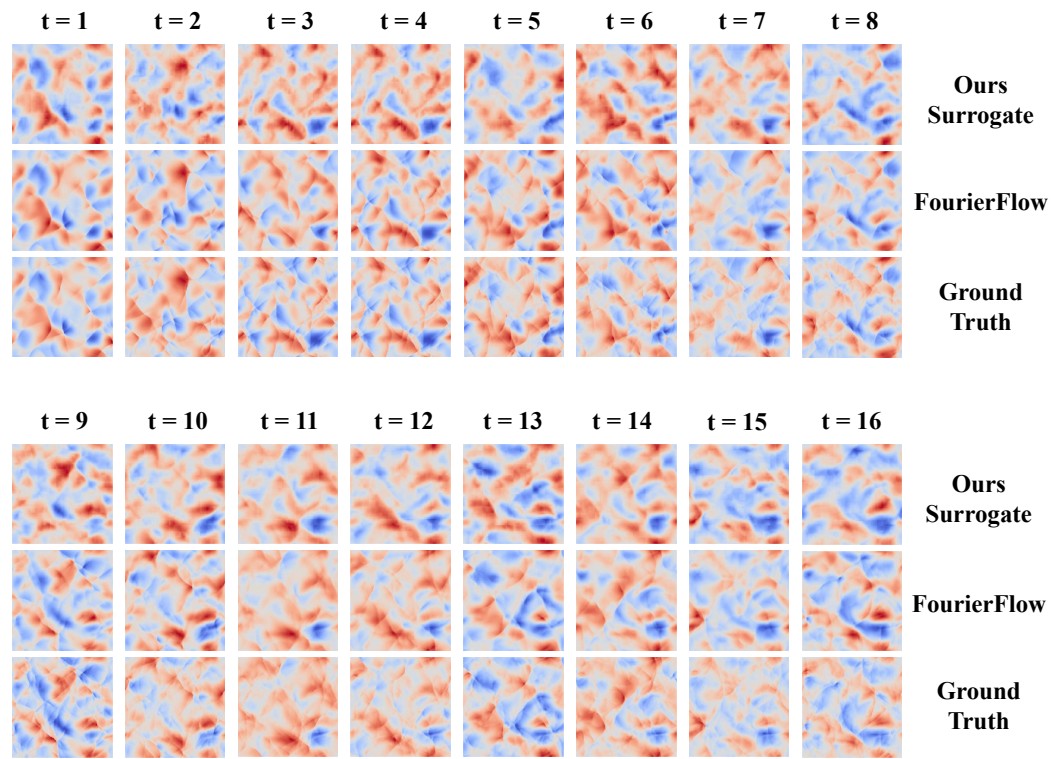

Figure 13: Visualization of the velocity-y fields produced by two methods for the compressible N-S.

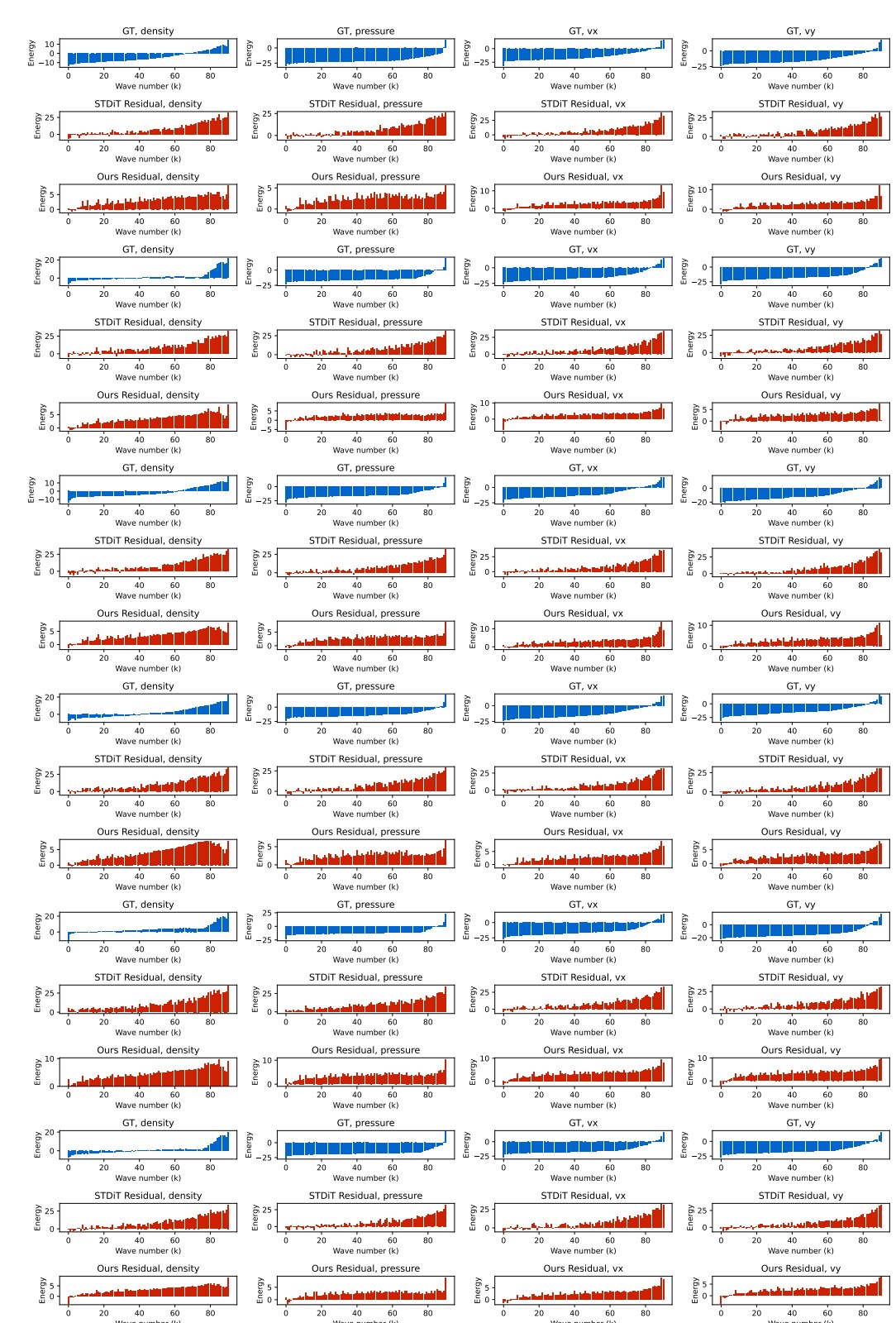

Figure 14: Visualization on wavenumber-energy trend.

