# OpenReview forum: "FourierFlow: Frequency-aware Flow Matching for Generative Turbulence Modeling"
_ICLR.cc/2026/Conference — Submitted to ICLR 2026_

### Official Review · Reviewer_SrwQ · 2025-10-19

**Soundness:** 2
**Presentation:** 3
**Contribution:** 3
**Rating:** 2
**Confidence:** 3

**Summary:**

The authors introduce a flow matching model to generate fluid flow simulations. Their model has two branches, one operating spatially and the other in the frequency space. They also use a pretrained masked autoencoder to improve the generation of high-frequency features.

**Strengths:**

- The authors deal specifically with the spectral aspects of fluid flows
- The experimental section investigates a variety of questions against many baselines

**Weaknesses:**

- The core contribution is about generative modeling for turbulence, but the paper only evaluates on 2D data where turbulence does not exist, see [1]. Furthermore, [1] is a closely related work that should be discussed.
- The samples shown in the appendix all have an unphysical 8x8 block structure

[1] Lienen, M., Lüdke, D., Hansen-Palmus, J., & Günnemann, S. (2023). From zero to turbulence: Generative modeling for 3d flow simulation.

**Questions:**

- How many steps did you roll out for Table 1?
- Line 160: Can you elaborate on this analysis that shows that all state-of-the-art generative models exhibit this?

---

### Official Review · Reviewer_4svf · 2025-10-27

**Soundness:** 2
**Presentation:** 2
**Contribution:** 2
**Rating:** 2
**Confidence:** 5

**Summary:**

FourierFlow is a frequency-aware generative framework designed to address the key challenges of spectral bias and common-mode noise that limit the fidelity of diffusion and flow-matching models in turbulence modeling.

The model enhances the representation of high-frequency turbulent structures—vortices, shocks, shear layers—while maintaining global coherence and physical consistency, outperforming existing state-of-the-art generative and surrogate PDE models.

**Strengths:**

The paper presents the first formal spectral-bias analysis of diffusion and flow-matching models in the context of physical PDEs.

**Weaknesses:**

Limited Physical Grounding Beyond Frequency Domain
Heavy Reliance on Learned Frequency Priors
Computational and Memory Overhead
Conceptual Ambiguity: Frequency is not from Physics, it is hidden

**Questions:**

Would a different diffusion process (e.g., cosine schedule) alter the bias?
How do these frequencies relate to physical length scales in turbulent flows?
How do we know FourierFlow’s high-frequency components correspond to real turbulence (not learned artifacts)?
why we need this? (the simulation for the case if using CFD is going to be cheap.)
What is the new knowledge in this paper?

What can we learn from the conclusion? "In this paper, we have introduced FourierFlow, a frequency-aware generative framework that addresses key limitations in turbulence modeling, such as spectral bias and common-mode noise. Through a novel dual-branch architecture and surrogate feature alignment, it achieves superior accuracy, physical consistency, and generalization across complex fluid dynamics tasks."

---

### Official Review · Reviewer_EYEq · 2025-10-28

**Soundness:** 3
**Presentation:** 3
**Contribution:** 2
**Rating:** 4
**Confidence:** 3

**Summary:**

FourierFlow proposes a new flow-based generative model for turbulence modelling. It builds upon three methods proposed in previous works to improve the modelling capacity with regards to high frequency features of turbulent flows.

**Strengths:**

- The paper is well written and experiments as well as appendix seem polished and extensive.
- Tackles a meaningful problem: high-frequency fidelity is crucial in turbulence modelling.
- Methodology is presented concisely and sound.
- Each component is motivated empirically through an ablation study.
- Provided code for reproducibility.

**Weaknesses:**

My main critique for the proposed method is the motivation and argumentation for the addition of three previously proposed components.

Intuitively I understand the problem of high frequency features. However, the actual arguments and motivations for specific components of the proposed method remain a bit inprecise to me. They mix arguments against diffusion models with arguments for some general components:

- Empirical spectral bias: It seems to me like your approach now fits lower frequency features worse. Not clear to me if we do not just trade-off low for high frequency components.
- Theoretic standpoint on spectral bias: Even though diffusion and in my opinion also flow matching models favour low and high frequency features at different "timepoints", it does not necessarily imply a problem. In fact, this can actually help the modelling of the different features, arguably a reason why they perform very well on many tasks. Note for example that within the modelling of complex systems there exist a lot of models which also explicitly separate those levels in hierarchical models.
- Further, the theoretic results are limited in generality. Theorem 4.1 is intuitive and essentially follows from assuming a power-law spectrum for x̂_0(ω) and that diffusion noise variance is frequency-independent. The novelty of this theorem is limited: the argument is not surprising and relies on strong assumptions (power law PSD) that may not hold uniformly across datasets. The authors should (i) explicitly state assumptions and the practical implications/limits and (ii) discuss whether/why flow-matching (deterministic) avoids the same fate (they use flow matching in the method but Theorem is about diffusion).
- Only arguing about energy preservation, but not showing or constraining it. Would be nice to actually constrain model to physical properties.
- Missing statistical significance test for results.
- The paper emphasizes that flow matching yields faster inference, but does not provide wall-clock comparisons (inference time), or ODE solver steps for flow matching vs iterative diffusion baselines.

Overall, my main concern is the actual motivation and novelty of the proposed method, where the main contribution is to combine 3 pre-existing components to show some improvement.

Minor points:
- Figure 1 different y scales make the figure unintuitive. Fixed scale would actually let their method stand out more.
- The paper is very crammed, even breaking the ICLR template (e.g. 5.3 section heading, a lot of negative v-space all over the paper).
- Space hacking let to figure 6 appearing after figure 8.

**Questions:**

- Could you report physics-preserving losses. Turbulence modelling values conservation/energy spectra, can you report spectral energy preservation metrics or discuss whether FourierFlow preserves invariants (kinetic energy, divergence-free constraints for incompressible flow)
- Can you show how your approach actually improves the spectral bias more explicitly?
- Explain to me the reason for a flow matching vs diffusion model. I.e., the components could be presented for many other methods.
- Why do the authors only compare to a surrogate model in the generalization experiment?
- The proposed MAE alignment and FM branch are somewhat heavy solutions. Why did you not consider other approaches to mitigate spectral bias (e.g., spectral weighting in loss, explicit high-frequency loss terms, or directly upweighting residual spectrum in training)?

---

### Official Review · Reviewer_9PxS · 2025-10-30

**Soundness:** 3
**Presentation:** 1
**Contribution:** 3
**Rating:** 2
**Confidence:** 4

**Summary:**

This paper has been submitted before and has been extensively provided with feedback. As the authors have chosen to resubmit this paper with only marginal changes (by pdfdiff analysis) I permit myself to recycle my old review. I apologise if this causes minor inconsistencies. But it seems inadequate to me that a paper that despite the work that previous referees have put into their reviews, essentially nothing has changed.

The authors propose an architecture for learning turbulent flow with data generated from numerical solutions of the Navier-Stokes equation as given by some examples picked from The Well and PDEbench. The authors here combine flow matching as a generative model with their architecture for the underlying neural network. The two key components are described as follows. On the one hand there is one branch that is based on the differential attention formalism proposed in [12]. In this branch there is a modeled separation of of low and high frequencies, called attention 1 and 2. The information from these branches is then added and weighted with a parameter. On the other hand, there is a fourier mixing branch network which has FNO-like layers where there is one weight transformation for all modes which is scaled up with some polynomial of the (norm of) the wave vector. Both streams are added up with a learnable, adaptive weighting. In their theoretical analysis, the authors claim there is a fundamental problem for diffusion architectures to reproduce small spatial details of fluid configurations. To substantiate this claim, the authors analyze the signal to noise ratio of the spatial image over the time of noising in the forward direction of a diffusion process. As the applied noise is 'spatially white' all modes are hit by noise in Fourier space of the same size. The authors then assume that the initial conditions fulfill a power law decay in frequency space, which is reasonable for turbulent flow. The signal to noise ratio for each mode can now be computed explicitly and it is shown that the suppressed - higher - spatial modes get a high signal to noise ration earlier. The authors propose, that their flow matching based approach with the one branch that can resolve high frequencies better due to the upscaling of weight sensitivity with the mode vector. While this is not supported with a theoretical analysis, the authors offer three empirical studies for the compressible Navier Stokes equation with different mach numbers. The main evaluation is on initial conditions from the training data set, while the generalization to unseen initial data is a separate study for different values of the viscosity. In their evaluation based on residuals (MSE) and the maximum error, the authors find in almost all cases an reduced MSE compared to various baselines. The authors also provide an ablation study on the effectiveness of their single modules.

**Strengths:**

+ The paper suggests a new architecture for flow matching (FM) based generation of solutions to the Navier Stokes equation
    + The numerical evaluation gives visually decent results and the comparison with a large number of other operator learning frameworks shows mostly superior results.
    + The code will be released in the case of acceptance making the results reproducible

**Weaknesses:**

My main criticism against the paper is the way it is prepared. It absolutely lacks precision and puts wordy descriptions in the place of clear scientific arguments supported by facts. To give some examples: There are words like "external representation space" in the section 3.3 where the meaning is never explained, in the context of physics the authors speak about "semantically meaningful [...] features" or the "common mode signal" which's Wikipaedia definition is clearly (completely) out of context, finally calling the rot operator spatial differential is unnecessarily complicated. I believe that this paper has a considerable and respectable technical core (see strengths). It is a pity that the preparation is so poor.
    Not unrelated, Theorem 1 and the supporting lemmas does not really contribute to the understanding why the model proposed by the authors should be more effective that a diffusion based generative model. The authors argue that in the diffusion noising, high frequencies modes are noised faster (if one accepts the spectral power law for the initial conditions). But that of course does not imply that these modes will be badely reconstructed during generation, but just that the backward process generates them last, which is just as it supposed to be. To make a theoretical problem out of this in my eyes is unfounded. The mathematical argument itself (not the interpretation) is ok and is based on standard arguments of the Itô calculus. The mathematical originality of this part is low.
    The notation in the theoretical part and Appendix F is very hard to follow. The authors should concisely define the spatial Fourier transform with which they work at least in the appendix and explain their discretization to FFT. Lemmas 1-3 are standard and stuff from stochastic analysis and should rather be in the appendix.
    In the evaluation, a residual based analysis in a chaotic setting over longer time spans makes little sense. It would be good if this evaluation could be complemented by comparing statistical flow features like power spectra of spatial and temporal correlations, which would be a good additional to the motivation of the paper.
    The most general version of PDE in the preliminary part is unnecessary, the explanation of the flow matching approach is largely incomprehensible and in many details inaccurate.

**Questions:**

* Figure 3 to me is very confusing. Where is the VideoViT in this figure - does the middle module represent the architecture of the learned FM vector field? Like the rest of the presentation, this figure should be cleaned up.
* Transformers work on a dimension reduced representation anyhow - why do you think that the high frequencies can't be inserted by the VAE part of something like stable diffusion? The GAN loss functions on small patches would be a natural place to improve small structures via a multiresolution approach like in pic2pixHD. Where do you see a serious problem in following this route?
* Why the generalization over initial conditions is not more in the focus? This is the real application case. How is the initial condition generated in this case differs from those in the training set. Are they just not included in the training set or are they generated with another statistical law?
* How would you generalize your approach to non periodic boundary conditions?

* Why didn't you work with the feedback from four (!) previous referees and rewrite your paper?

---

### Meta-Review · Area_Chair_RKei · 2026-01-05

**Summary:**

The authors introduce a generative model for learning turbulent flows from simulations. They first empirically highlight the weaknesses of existing generative models in capturing high-frequency regions of the PDE domain. They then propose an architecture that combines flow matching with dedicated components designed to alleviate the spectral bias commonly observed in generative models. The proposed approach integrates several components from the literature in a novel manner. Theoretical results analyze how frequency components evolve within a diffusion process. Experiments are conducted on two fluid flow datasets and demonstrate improvements over selected baselines.

The reviewers acknowledge the relevance of the problem and the empirical improvements reported in the evaluation. However, they highlight several weaknesses, including a lack of precision in the technical description, insufficient motivation and explanation for the choice and combination of the model components, and the fact that the theoretical analysis does not adequately support the paper’s claims.

**Reviewer Concerns:**

The authors chose not to address the reviewers’ remarks and questions.

**Reviewer Scores:**

Nio rebuttal provided

---

### Decision · Program_Chairs · 2026-01-26

Reject